



# Volatile organic compound emissions from solvent- and water-borne coatings: compositional differences and tracer compound identifications

Chelsea E. Stockwell[1,2], Matthew M. Coggon[1,2], Georgios I. Gkatzelis[1,2,a], John Ortega[1,2], Brian C. McDonald[1], Jeff Peischl[1,2], Kenneth Aikin[1,2], Jessica B. Gilman[1], Michael Trainer[1], Carsten Warneke[1,2]

[1] NOAA, Chemical Sciences Laboratory, Boulder, Colorado 80305, United States

[2] Cooperative Institute for Research in Environmental Sciences, University of Colorado, Boulder, Colorado 80309, United States

[a] now at: Institute of Energy and Climate Research, IEK-8: Troposphere, Forschungszentrum Jülich GmbH, Jülich, Germany

**Correspondence**: Chelsea Stockwell (chelsea.stockwell@noaa.gov) and Carsten Warneke (carsten.warneke@noaa.gov)

**Abstract.** The emissions of volatile organic compounds (VOCs) from volatile chemical products (VCPs) - specifically personal care products, cleaning agents, coatings, adhesives, and pesticides - are emerging as the largest source of petroleum-derived organic carbon in US cities. Previous work has shown that the ambient concentration of markers for most VCP categories correlate strongly with population density except for VOCs predominantly originating from solvent- and water-borne coatings (e.g., parachlorobenzotrifluoride (PCBTF) and Texanol®, respectively). Instead, these enhancements were dominated by distinct emission events likely driven by industrial usage patterns, such as construction activity. In this work, the headspace of a variety of coating products was analyzed using a proton-transfer-reaction time-of-flight mass spectrometer (PTR-ToF-MS) and a gas chromatography (GC) pre-separation front-end to identify composition differences for various coating types (e.g., paints, primers, sealers and stains). Evaporation experiments of several products showed high initial VOC emission rates and for the length of these experiments, the majority of the VOC mass was emitted during the first few hours following application. The percentage of mass emitted as measured VOCs (< 1 to 83%) mirrored the VOC content reported by the manufacturer (<5 to 550 g L$^{-1}$). Ambient and laboratory measurements, usage trends, and ingredients compiled from architectural coatings surveys show both PCBTF and Texanol account for ~10% of the total VOC ingredient sales and therefore can be useful tracers for solvent- and water-borne coatings.

## 1 Introduction

After decades of declining mixing ratios of volatile organic compounds (VOCs) in urban areas (Warneke et al., 2012), volatile chemical products (VCPs = coatings, adhesives, inks, personal care products, and cleaning agents) have emerged as a major source of petrochemical organics in the urban atmosphere (McDonald et al., 2018a). Measurement and modeling efforts have already shown that VCP emissions and their contribution to urban ozone formation are comparable to those for fossil fuel VOCs in Los Angeles and other cities in the United States (McDonald et al., 2018a;



Coggon et al., 2020). Reductions in tailpipe emissions of VOCs from gasoline vehicles are slowing, and diminishing
returns from emission control technologies have been reported for on-road vehicles (Bishop and Haugen, 2018). With
slowing trends in ozone precursors and shifts in ozone production regimes, decreases in ozone design values may have
slowed as well (Parrish et al., 2017).

A fuel-based inventory by McDonald et al. (2018a) showed that about 18% of the petrochemical VOC

emissions in Los Angeles in 2010 were from personal care products. Coggon et al. (2018) found that D5 siloxane,
which is a unique tracer for personal care product emissions, is emitted in urban areas in similar amounts as benzene
from vehicles, indicating a significant emission source of personal care products. The next largest emission source
was from coatings accounting for approximately 13% of the VOC inventory in Los Angeles. Coatings in emission
inventories are defined as paints, varnishes, primers, stains, sealers, lacquers, and other solvents associated with
coatings (e.g., thinners, cleaners, additives). This includes both industrial and architectural uses, which comprise half-
and-half of the coating fraction. The impact of VOC emissions from coatings has been investigated in the indoor
environment (Corsi and Lin, 2009; Weschler and Nazaroff, 2012; Schieweck and Bock, 2015; Kozicki et al., 2018),
but few measurements have been reported in the outdoor environment. Urban air is likely impacted from coating
emissions from indoor/outdoor exchange as well as from architectural and industrial coating usage outdoors. To
quantify VCP coating emissions in urban air, it is important to identify VOC tracers that are uniquely linked to water-
and solvent-borne usage, analogous to the emissions of D5 siloxane from personal care products. Although there is a
lack of detailed analysis of coatings emissions in the ambient atmosphere, emissions suspected from coatings have
been measured in ambient air. Goliff et al. (2012) measured 2,2,4-Trimethyl-1,3-Pentanediol Monoisobutyrate (TPM,
also known as Texanol®), a common solvent in water-borne coatings, in Southern California at mixing ratios of up to
20 ppt with the largest values in summer, when coating activities are typically the highest.

The composition and emissions of chemical products have changed significantly in recent decades in an

effort to reduce the ozone ($O_3$) and secondary organic aerosol (SOA) formation potential (Weschler, 2009; CARB,
2015; Shin et al., 2016). For example, water-borne paints are increasingly replacing solvent-borne paints, while
simultaneously many VOC ingredients are being replaced with water (Matheson, 2002), "exempt" VOCs, or low-
vapor-pressure VOCs (LVP-VOCs) (Li et al., 2018). The California Air Resources Board (CARB) defines "exempt"
VOCs as compounds that are not expected to meaningfully contribute to ozone formation due to their low reactivity
in the atmosphere. Examples include acetone, ethane, perchloroethylene, methyl acetate, and
parachlorobenzotrifluoride (PCBTF). CARB also exempts several LVP-VOC, which are defined as chemical
compounds containing at least one carbon atom and a vapor pressure less than 0.1 mm Hg at 20°C, organic compounds
with more than 12 carbon atoms, compounds with boiling points greater than 216°C, chemical mixtures comprised
solely of compounds with more than 12 carbon atoms, or as the weight percent of a chemical mixture that boils above
216°C (Li et al., 2018). As a result of product reformulations and VOC exemptions, many paints can now be classified
as "zero VOC" paints that fall below the government-regulated permissible amounts, even though they emit
compounds that are more broadly defined within the category of VOC.

Presented here are ambient and laboratory measurements of vapors emitted from coatings using a proton-

transfer-reaction time-of-flight mass spectrometry (PTR-ToF-MS). Ambient measurements show the spatial and



temporal trends of PCBTF and Texanol emissions in New York City (NYC), Chicago, Pittsburgh, and Denver. These
ambient measurements are linked to coatings using a series of laboratory measurements sampling various architectural
coatings. First, we compare VOC composition from coating headspace samples analyzed using PTR-ToF-MS with
gas chromatograph (GC) pre-separation. These measurements highlight the differences in VOC composition for a
variety of coating products and help to unambiguously identify VOC tracers linked to solvent and water-borne
coatings. Emission factors (g kg$^{-1}$) of VOC mass are measured via controlled evaporation experiments to quantify key
VOC emissions from coating use. Lastly, ingredient compilations from architectural coatings surveys from CARB are
compared with the laboratory measurements to confirm usage trends. Coating surveys generally agree with the
ambient and laboratory measurements and support the assignment of Texanol and PCBTF as atmospheric tracers for
water-borne and solvent-borne coatings.

**2 Experimental Methods**

**2.1. Instrumentation**
**2.1.1 PTR-ToF-MS**
Mixing ratios of VOCs in ambient and laboratory measurements were determined using a proton-transfer-reaction
time-of-flight mass spectrometer (PTR-ToF-MS; referred to hereafter as PTR-ToF) (Yuan et al., 2016; Yuan et al.,
2017). The PTR-ToF measures a large range of aromatics, alkenes, nitrogen-containing species, and oxygenated
VOCs. Instrument backgrounds were determined for laboratory measurements before and after every experiment for
short duration experiments, or every 2 h for longer duration experiments and ambient measurements, by passing air
through a platinum catalyst heated to 350°C. Data were processed following the recommendations of Stark et al.
(2015) using the Tofware package in Igor Pro (WaveMetrics). The PTR-ToF was calibrated using gravimetrically-
prepared gas standards for typical water-borne solvents such as acetone and methyl ethyl ketone, and solvent-borne
compounds such as toluene and C8-aromatics. Texanol and PCBTF were calibrated by liquid calibration methods as
described by Coggon et al. (2018). The sensitivities for Texanol and PCBTF for 1 second measurements was 9 and
69 normalized counts per second per ppbv (ncps ppbv$^{-1}$), respectively, and the detection limits for both were < 10 pptv
(Gkatzelis et al. 2020a). The sensitivity of compounds that were not calibrated were calculated according to Sekimoto
et al. (2017).

The PTR-ToF is less sensitive to smaller hydrocarbons (de Gouw and Warneke, 2007; Sekimoto et al., 2017)

and substantially underestimates their mixing ratios. PTR-ToF sensitivities to alkanes and alkenes are typically only
a few percent of those for oxygenates (Warneke et al., 2003), and the sensitivities calculated using methods outlined
in (Sekimoto et al., 2017) are likely overestimated. Theoretical calibration factors from reaction rate coefficients have
been shown to be biased high for alkanes and alkenes (Warneke et al., 2003). In order to more accurately quantify the
contributions of hydrocarbons (HCs), the calculated sensitivities for low molecular weight HCs (< C5) were assumed
to have the same sensitivity as low molecular weight alkanes (10 ncps ppbv$^{-1}$). This results in an estimated uncertainty
in total VOC emissions of about a factor of two.



### 2.1.2 GC Front End to PTR-ToF-MS

PTR-ToF-MS only resolves VOC molecular formulae. Gas chromatography (GC) pre-separation has been used previously to identify structural isomers (Warneke et al., 2003; Koss et al., 2016). Here a custom-built GC described by Kuster et al. (2004) was updated and re-designed specifically as a PTR-ToF-MS front end to analyze the complex headspace mixture of select coating formulations. Details describing the setup and performance of the GC interface are provided in the supplemental information and only a brief description is provided here.

The GC consists of a 30 m DB-624 column (Agilent Technologies, 30 m, 0.25 mm ID, 1.4 µm film thickness) and oven combination identical to the system described by Lerner et al. (2017), a liquid nitrogen cryotrap, and a 2 position 10-port valve (VICI) to direct gas flows. The column was selected to measure polar and nonpolar VOCs in the approximate range of C4-C10. The effluent of the GC column is injected into the PTR-ToF inlet. Depending on the application, 1-5 minute samples can be collected and chromatogram lengths of 10-20 minutes can be chosen such that the total run time is between 15 and 30 minutes. LabVIEW (National Instruments) software controls the sequence of events, hot and cold trap temperatures, valve switching, sample flow and carrier gas flow. The detection limit for commonly detected VOCs (e.g., isoprene, benzene, xylenes) using this cryofocusing system is ~ 5 pptv.

### 2.2. Laboratory Measurements

Laboratory measurements were performed to qualitatively evaluate the headspace VOCs emitted from commonly used coatings. Nineteen different solvent- or water-borne formulas were tested, and ranged in applications including paints, stains, primers, sealers, and preservatives. It is worth noting that industrial coatings formulated to withstand extreme environmental conditions were not a part of this survey. The headspace VOCs were sampled by placing the product container close to the PTR-ToF inlet and zero air was sampled between runs. The GC-front end was used to aid the specific identification of VOC isomers emitted from different coating types, which were compared to the ingredient lists or CARB coating surveys.

Several representative products were sampled over longer periods to investigate their evaporative behavior: a solvent-borne polyurethane stain, a latex paint, a primer/sealer paint, and a "zero VOC" low odor paint. These evaporation experiments were performed by flowing synthetic air (2 L min$^{-1}$) through a Teflon coated chamber enclosing a microbalance scale (Ohaus) for a minimum of 19 hours. Prior to each experiment, the chamber was flushed until background VOC concentration were < 50 ppt. Experiments were initialized by depositing small (< 50 mg) samples of coating product through a septum and onto a small piece of Teflon plastic placed on the scale. The PTR-ToF monitored VOCs from the exhaust of the Teflon chamber while changes to the product mass were recorded by the scale. VOC emission factors (g kg$^{-1}$) were determined by dividing the integrated VOC signal measured by PTR-ToF over the course of the experiment by the initial mass on the scale and the percentage of mass emitted as VOCs is determined by dividing by the total change in mass on the scale.

### 2.3. Mobile and Ground-site Measurements





The field measurements used to evaluate ambient measurements of coatings are described in detail by Coggon et al.
(2020) and Gkatzelis et al. (2020a). Briefly, measurements were conducted during the New York Investigation of
Consumer Emissions (NY-ICE) campaign in winter (March 5 – 28, 2018) and the Long Island Sound Tropospheric
Ozone Study (LISTOS) in summer (July 5 – 24, 2018) to characterize the emission profile, strength, and seasonality
of VOCs emitted from VCPs.
Ground-site measurements were performed at the City College of New York (CCNY) campus. Mobile VOC
measurements were conducted throughout NYC and other urban regions (Pittsburgh, Chicago, and Denver) using the
NOAA Chemical Sciences Laboratory (CSL) mobile laboratory to characterize the spatial distribution of VCP
emissions. The mobile laboratory was driven through the boroughs of New York City, Long Island, and eastern New
Jersey, and multiple drives were performed in Chicago (2) and Denver (3). In each case, the mobile laboratory was
driven downwind, upwind, and through the city center to evaluate urban VOC enhancements. Drive routes were
selected to sample regions of both high and low population density to investigate differences in VCP emissions.

**2.4. FIVE-VCP Emission Inventory**
The Fuel-based Inventory of Vehicle Emissions and Volatile Chemical Products (FIVE-VCP) emission inventory used
in this work was described in detail by Coggon et al. (2020). Briefly, VCP emissions were estimated from chemical
production data determined from a "bottom up" mass balance of the petrochemical industry. The VOC speciation
profiles were updated for the coatings category to include more recent architectural coating surveys by the California
Air Resources Board (CARB) (CARB, 2018). The per capita use of VCPs specific to the coating sector was estimated
and combined with VOC emission factors (in g VOC per kg product) reported by McDonald et al. (2018a) that is
based on a review of indoor air quality literature. The VCP emissions were spatially apportioned using US Census
block population data (Bureau USC, 2020a) and the temporal patterns are taken from the National Emissions Inventory
(NEI) 2014 (EPA, 2017).
Mobile source emissions are estimated utilizing a fuel-based approach based on fuel sale reports for on-road
and off-road engines. CO and VOC emission factors (in g VOC per kg fuel) were taken from the compilation by
McDonald et al. (2018a) and updated to 2018 by Coggon et al. (2020). The emission factors include tailpipe emissions
from running exhaust, enhanced emissions associated with cold-starting engines, and evaporative gasoline VOC
sources. The spatial and temporal emission patterns of mobile source engines are taken from the NEI 2014 (EPA,
2017).

**3 Ambient Measurements of PCBTF and Texanol**
The NYC ambient measurements of PCBTF and Texanol from the ground site and the mobile laboratory in 2018 are
summarized in Fig. 1. In Fig.1a, the drive tracks overlay a map of the population density in the region and are color
coded with summer-time PCBTF and Texanol. The population density is highest in Manhattan with more than 28,000
people km$^{-2}$. Generally, the highest mixing ratios of PCBTF and Texanol were found in the areas with the highest
population density, but were mostly dominated by short, spatially isolated plumes. PCBTF and Texanol were binned
along the east-west transects together with the population density as shown in the top panel. The correlations with





population density for both compounds ($R^2$= 0.23 and 0.57) were lower than what was found for most other VCP
tracer compounds ($R^2$ > 0.8, Gkatzelis et al., 2020a). This is consistent with PCBTF and Texanol being from
construction activities, rather than correlated with population like other VCP tracers (e.g. D5 siloxane for personal
care products). In the lower panel, the time series from the ground site measurements in winter and summer for PCBTF
and Texanol are shown together with CO, a combustion tracer, and D5 siloxane, a personal care product tracer
(Coggon et al., 2018). At the end of the winter ground-site measurements in late March, a stagnation period, marked
in Fig. 1b with the orange box, increased the mixing ratios of all VOCs and CO. The correlations with CO for the
winter measurements are also shown in Fig. 1c. Throughout the entire campaign, the correlation of PCBTF and
Texanol with CO is low, except for the time period at the end of March (orange) where the urban emissions of near-
by sources accumulated at the measurement site. PCBTF and Texanol are also poorly correlated with D5 siloxane,
except during the stagnation period ($R^2$= 0.50 and $R^2$=0.76). This suggests that PCBTF and Texanol have sources
other than combustion and personal care products and the large spikes in PCBTF observed during the drives shows
that high emissions are from distinct point sources, and not from dispersed sources like traffic.

Even though PCBTF and Texanol have different sources than CO, the correlation with CO during the

stagnation event can be used to estimate the PCBTF and Texanol emissions in NYC following the method described
by Coggon et al. (2020), which uses the slope from the correlation together with CO emissions from the bottom-up
FIVE mobile source inventory (McDonald et al., 2018b). The resulting emissions for PCBTF and Texanol in NYC
were approximately 1 and 0.1 mg person$^{-1}$ day$^{-1}$, respectively.

Figure 2a summarizes the mobile laboratory measurements in Chicago, Pittsburgh, Denver, and the transit in

between the cities, where the drive tracks close to the downtown areas in each city are color coded by PCBTF mixing
ratio and plotted on top of the population density. Like NYC, PCBTF, and Texanol were generally enhanced in urban
regions, but not well correlated with urban population density. PCBTF was significantly enhanced in Chicago in
distinct locations, and not well correlated with D5 siloxane (Fig. 2b, $R^2$ < 0.1). The population density dependence of
PCBTF and Texanol for winter and summer is shown in Fig. 3a as the ratio of the respective tracer with benzene,
which is used as a tracer for mobile source emissions. This ratio controls for meteorology between cities, and also
reflects differences in the proportion of VCP and traffic emissions across urban regions. As population density
increases, mobile source emissions plateau because of roadway capacity and increased mass transit usage (Gately et
al., 2015), while emissions from VCPs scale linearly with the number of people. Consequently, the ratio of a VCP
markers (e.g. D5 siloxane) with benzene is higher in more densely populated regions (Gkatzelis et al., 2020a). PCBTF
and Texanol do not exhibit a population density dependence ($R^2$ < 0.2), especially compared to D5 siloxane ($R^2$=0.82)
(Fig. 1 in Gkatzelis et al. (2020a)), which likely indicates that emissions from coatings are influenced by industrial
usage patterns, such as construction activity, rather than consumer product usage patterns, which impact the spatial
and temporal variability of D5 siloxane (Coggon et al., 2018; Coggon et al., 2018; Gkatzelis et al., 2020a).

The weekly profiles of the PCBTF and Texanol ratios for summer and winter NYC ground site

measurements are shown in Fig. 3b. Winter ratios are lower than those in summer, which likely indicates a seasonal
change in emissions due to relatively fewer coating projects in the wintertime. The same was observed in the other
cities as can be seen in Fig. 3a, where the winter data show a much smaller correlation with the population density





compared to summer. In addition to usage trends, it is also likely meteorology accounts for some of changes in
emissions between seasons and further complicates interpretation. Texanol ($C_{12}H_{24}O_3$) is detected as a fragment at $m/z$
199 ($C_{12}H_{22}O_2$). It has been suggested that compounds formed via chemical processes are a potential interference at
that mass during summer months (Gkatzelis et al., 2020a). A PMF analysis of the ambient dataset used herein is
described in detail in Gkatzelis et al. (2020b) and noted that the majority of $m/z$ 199 was attributed to the VCP
emissions in summer NYC (>60%), while the remaining fraction was attributed to daytime/morning chemical
processes influenced by VOC oxidation. Therefore, photochemistry might contribute to the larger Texanol ratio
observed in summer compared to winter and complicates the use of PTR-ToF measured Texanol as a tracer for paints
during summer months. Nevertheless, Goliff et al. (2012) observed the highest mean ambient concentrations of
Texanol from sorbent-tube collection during summer months in multiple cities, which are unaffected by interferences
to other compounds. It might also be expected that coatings emissions are smaller on the weekends compared to
weekdays, but this difference is not clearly observed in Fig. 3b. Currently, the FIVE-VCP does not take the seasonality
of VCP emissions into account and therefore likely overestimate coatings emissions during winter.

**4 Headspace and Evaporation Measurements**
Gkatzelis et al. (2020a) used the population density dependence of various VOCs to identify specific tracers for VCP
emissions. Although PCBTF and Texanol ambient mixing ratios did not show a strong population dependence, they
met additional criteria for selection as VCP tracers, i.e., they were regularly measured at significant mixing ratios in
ambient air and are uniquely represented in the FIVE-VCP emission inventory as a component of coatings. To
demonstrate the prevalence and efficacy of these compounds as markers in the coating VCP category, laboratory
measurements were performed to sample VOCs from the headspaces of commonly used architectural coatings and to
quantify the fraction of PCBTF and Texanol associated with the measured VOC mass.

To understand the composition of the emissions from coating products in greater detail, the headspace of

nineteen different coatings were measured by PTR-ToF. To capture the two ends of the spectrum, both a water-borne
"zero VOC" low odor paint and a solvent-borne polyurethane stain were tested with the GC-front end pre-separation
method described above (GC-PTR-ToF-MS). The GC-PTR-ToF-MS chromatograms of select (a) small oxygenates,
(b) hydrocarbons (HC), and (c) aromatics are shown in Fig. 4. For the "zero VOC" paint, the main oxygenate emitted
was acetone, with smaller emissions from other solvents including methanol, ethanol, and methylethylketone (MEK).
The compounds identified in the "zero VOC" GC hydrocarbon panel are primarily fragments of oxygenated
compounds and, generally, oxygenated compounds were the dominant emission from "zero VOC" paint. Emissions
from aromatics were negligible. The mass for Texanol and its fragments were observed in the direct PTR-ToF
sampling of the headspace of several water-borne products, but pure Texanol injections into the GC-PTR-ToF-MS
demonstrated that Texanol does not elute from the GC column either because it is lost to the water trap or has too low
of a vapor pressure to elute from the column under the selected temperature program.

The GC peaks of the solvent-borne polyurethane in the chromatograms were several orders of magnitude

larger than those for the water-borne product. Acetone and MEK were the most abundant small oxygenates, while
ethanol and methanol emissions were negligible. Over 30 distinct hydrocarbon peaks were detected with retention



259 times over 300s and with mass 55 ($C_4H_6 \cdot H^+$) and mass 69 ($C_5H_8 \cdot H^+$) dominating, which indicates C4- and larger

260 hydrocarbons as the primary constituents. Stoddard solvent is commonly listed as an ingredient in solvent-borne

261 coatings, and mainly consists of < C10 alkanes, cycloalkanes, and aromatics (Censullo et al., 2002), which are clearly

262 evident in the mass spectrum. These hydrocarbons are likely not useful unique tracers for coatings products due to

263 overlapping contributions from gasoline and diesel fuel emission in urban areas (Gentner et al., 2012; Gentner et al.,

264 2013), and therefore a detailed discussion of all the hydrocarbon peaks is beyond the scope of this work. Benzene (*m/z*

265 79) is one of the aromatics with minor emissions, but most other C7-C12 aromatics including toluene (*m/z* 93) were

266 more significant. Benzene is often reduced in coating formulations due to its toxicity. The largest aromatic peak is

267 PCBTF, which is completely absent in all water-borne products that were tested.

268  Figure 5a shows PTR-ToF-MS mass spectra during the evaporation experiments of a "zero VOC" low odor

269 paint, primer/sealer paint, a latex paint, and a solvent-borne polyurethane stain. During the evaporation experiments,

270 the products were sampled over longer periods to investigate changes in evaporative emissions. The mass spectra were

271 taken when highly volatile compounds were evaporating early, following initial application of the paint onto the

272 balance, and later when most of the volatile compounds had already evaporated (beyond 3 hours). The mass spectra

273 are given in parts per billion (ppbv) and normalized to the weight loss of the product on the scale in the evaporation

274 chamber. The major peaks in each product are labeled by the most likely compound identification, as determined from

275 fragmentation patterns, headspace analysis, and GC-separation. The sums of all the measured PTR-ToF signals in the

276 "early" mass scan are identified by the labels for all measured VOCs, which includes regulatory-exempt species such

277 as acetone. The pie charts in Fig. 5b show the distribution of VOC mass emitted during each complete evaporation

278 experiment with the largest 10 species labeled.

279  The "zero VOC", low odor paint and the primer/sealer paint have relatively low emissions, which were

280 dominated by small oxygenated VOCs (methanol, ethanol, formaldehyde, and acetone). The primer/sealer paint also

281 emitted Texanol, which can be expected from the ingredients of many water-borne products. The Texanol signal

282 measured in the laboratory experiments has no contribution from the oxidation products that were identified as

283 potential interferences during ambient summer sampling (Gkatzelis et al., 2020a; Gkatzelis et al., 2020b). Methanol,

284 ethanol, and acetone emissions are expected ingredients, as listed in the FIVE-VCP inventory. Formaldehyde is not

285 an ingredient, but emissions might be expected from the use of preservatives that include formaldehyde-condensate

286 compounds that rely on the release of free formaldehyde from the hydrolysis of a parent structure such as dimethylol

287 glycol and dimethylol urea. Dimethylol glycol and dimethylol urea are routinely used as biocides in water-borne paints

288 and fungicidal products, but have now been widely replaced by other compounds such as isothiazolinones

289 (Salthammer et al., 2010). Ethylene glycol is an abundant ingredient in architectural coatings sales surveys (CARB,

290 2018), and has been observed in an indoor air study during a painting event (Pagonis et al., 2019). The signal at mass

291 45 ($C_2H_4O \cdot H^+$) is typically attributed in PTR-MS studies to acetaldehyde (de Gouw and Warneke, 2007), but ethylene

292 glycol fragments mostly to *m/z* 45. Acetaldehyde was present in the GC experiments and although ethylene glycol

293 was not observed, as it is likely lost in the water trap, its dominance in architectural coatings and paints suggests it

294 may be a dominant species at that mass.



The total VOC signal was significantly larger by an order of magnitude for water-borne latex paint than for
the two other water-borne paint products, with ethanol being the biggest emission. In addition to the small oxygenated
VOCs, larger compounds such as dipropylene glycol monobutylether and other ketones and acetates were emitted.
Several species listed as ingredients in CARB coatings surveys were tentatively identified in the paint emissions as
ethylene glycol, methyl-n-amyl ketone, 1-methyl-2-pyrrolidinone, and methyl-, vinyl-, ethyl-, and butyl acetate. For
butyl acetate, the major peak in the GC chromatogram at mass 117 correlated strongly with peaks at $m/z$ 61 and 43,
which are known fragments of the parent ion (Buhr et al., 2002).
As was expected from the GC analysis, the VOC emissions of the solvent-borne polyurethane were markedly
different and ~5 times larger than the water-borne paint. Hydrocarbons and aromatics are clearly evident in the mass
spectrum. Small masses (C<5) are known to be affected by fragmentation, which adds to quantification uncertainty,
and most hydrocarbon masses had multiple peaks in the GC chromatograms. As was noted in the headspace GC
experiments, the solvent-borne polyurethane also emitted PCBTF.
The weight loss recorded on the scale during the evaporation experiments equals the mass emitted as VOCs,
water (which was not measured), and other compounds undetectable by PTR-ToF. Each measured VOC (mg m$^{-3}$) was
converted to an emission rate (mg s$^{-1}$) by multiplying with the gas flow rate and integrated across the length of the
experiment to determine the total mass emitted by each individual species. Figure 6 shows the time series of mass
emitted as the sum of all measured VOCs overlaid with the weight change measured on the Teflon covered scale for
each evaporation experiment.
For the water-borne products in Fig. 6a-c, the mass emitted as VOCs is a very small fraction of the mass lost
from the coatings product (<1 to 7%) and the main weight loss is attributed to water. Fig. 6b shows the time series for
the primer/sealer paint, which was the product with the largest difference in VOC emission rate and mass lost. This
mass loss rate gradually leveled off, while the total VOC emission continued to increase at a steady rate. This was
largely driven by only a few compounds (e.g. ethylene glycol), and therefore the estimated mass loss and emission
factors reported herein represent lower-end estimates. Latex paint showed a similar behavior with VOCs steadily
rising even at the conclusion of the experiment. These results are not unexpected, as emissions of Texanol from paints
have been observed for periods as long as 15 months (Lin and Corsi, 2007) and these experiments were stopped after
19 hours. The VOC mass emitted, as detected by PTR-ToF, is related to the VOC content of the product, where the
latex paint clearly has the largest VOC emission and the "zero VOC" paint the lowest.
For the solvent-borne polyurethane coating in Fig. 6d, the mass emitted as VOCs exhibited a temporal profile
that more closely mirrored the mass lost recorded by the scale. The similarity in the emission profile and weight loss
indicate that the majority of the weight loss from the solvent-borne stain is due to VOC emissions with only a small
fraction of weight loss from other undetected compounds or underestimation due to poor sensitivity to hydrocarbons.
The mass remaining on the scale indicated that about 40% of the weight remained as solids or unevaporated VOCs,
which is within 5% of what is expected from the manufacturer reported VOC content of 550 g L$^{-1}$. Figures 6b and 6d
represent two extremes and it is likely each product will have different evaporative properties depending on the overall
composition and atmospheric conditions. Figure 6 also shows that for the (a) "zero VOC" paint and (d) polyurethane
stain, the VOCs emitted in the first hour account for over 50% of the total mass emitted as measured VOCs over the





course of this experiment. Latex paint (c) takes an additional hour and the primer/sealer (b) takes nearly 6 hours for the majority of the measured VOCs to evaporate. The experiment did not complete to dryness, thus the total mass measured is likely an underestimate, and potentially under-account semi- and intermediate-volatility VOCs that have a lower volatility. Continued emissions from dry-paint have been observed following complete water evaporation (Clausen, et al., 1991; Hodgson et al., 2000), though emissions from the dry-film were not investigated here.

Figure 7a shows the time series of the emission rates of the sum of all measured VOCs for each product (mg h$^{-1}$). The solvent-borne polyurethane had a maximum VOC emission rate (6.7 mg h$^{-1}$) that was more than a factor of two greater than the highest-emitting water-borne coating. For each product, the highest total VOC emission rate peaked within the first 6 minutes. The elapsed time to reach the maximum emission rate varied to within a few minutes, and this likely reflects the different volatilities of the ingredients. For the coatings tested here, the majority of the measured VOC mass is emitted within the first few hours, and therefore the most significant atmospheric implications for ozone formation likely occur during and shortly following application.

Figure 7b shows the relative contribution of select VOCs to the total VOC mass as a function of time during the evaporation of the primer/sealer paint. Emissions of methanol and 1-butanol fragments dominated at the start of the experiment, followed by other species including ethylene glycol and Texanol, which all peaked in a sequence that paralleled reported saturation vapor concentrations (NIST Chemistry WebBook). The volatilities of candidate species at mass 45 suggest ethylene glycol ($C_0 \sim 10^5$ µg m$^{-3}$) is the dominant compound, since the emissions of acetaldehyde ($C_0 \sim 10^9$ µg m$^{-3}$) would have peaked before methanol ($C_0 \sim 10^8$ µg m$^{-3}$). The emissions of acetone and formaldehyde are more complicated. Although the reported saturation vapor pressures are higher than methanol the maximum emission rate peaked later and changed more gradually, resulting in more prolonged emissions. It is clear that the contribution of specific compounds to the total mass emitted can vary significantly over time and likely depends on volatility ($C_0$), the initial composition of the product, and other factors such as drying time, temperature, humidity, and substrate interaction/properties. Texanol has a lower saturation vapor concentration ($10^5$ µg m$^{-3}$), and the maximum emission rate occurred ~11 minutes into the experiment, but the amount of mass emitted as Texanol during the first hour only accounted for 34% of the total Texanol emitted during the entire experiment (~19 h). It took over 6 hours to account for 75% of the total emitted Texanol, demonstrating that certain species can emit considerably across several hours to days. The majority of the Texanol emission does occur within the first several hours, and the fast evaporation supports its use as a tracer. Lin and Corsi (2007) showed emissions of Texanol decreased by 90% within the first 100 h following paint application, and mass closure assessments showed that airborne emissions of Texanol were greater than recovery from material components for thin-film flat paints. The VOC speciation in each evaporation experiment is shown in Fig. 5b.

The PTR-ToF also detects several inorganic species such as ammonia, though the signal at $m/z$ 18 suffers from high background signal and has only been quantified for large emission sources such as biomass burning (Karl et al., 2007; Müller et al., 2014; Koss et al., 2018). There was a clear enhancement of ammonia in all water-borne coatings that was absent in the solvent-borne coatings, and it is likely mass emitted as ammonia can be important in water-borne products as it is commonly used as a pH stabilizer. The ammonia PTR-ToF sensitivity derived from a comparison with an FTIR during biomass burning sampling (Koss et al., 2018) was used to estimate the mass emitted



as ammonia for the latex paint. The total mass emitted as ammonia (2.1 mg) rivaled the VOC mass (2.3 mg), and
shows ammonia emissions can be significant from certain water-borne products. There was no direct calibration of
ammonia during these experiments and therefore the discussion focuses on VOCs only.
Table 1 summarizes the results of the evaporation experiments. As expected, the solvent-borne polyurethane
VOC emissions accounted for the greatest amount of total mass detected (83%), which is likely a lower limit estimate
since the PTR-ToF is less sensitive to hydrocarbons. The trend in the VOC emissions tracks the VOC content reported
on the product labels (g L$^{-1}$). The "zero VOC" paint data sheet reported < 5 g L$^{-1}$ VOCs and emitted very few VOCs
with an emission factor (in g VOC emitted per initial weight of the product) of 0.7 g VOC kg$^{-1}$ paint, followed by the
primer/sealer (2.8 g kg$^{-1}$), latex paint (43.1 g kg$^{-1}$), and finally polyurethane (495 g kg$^{-1}$) with a reported VOC content
of < 550 g L$^{-1}$. The VOC emission rates for each product, averaged during the 19-hour experiments, also mirror the
VOC content. We note that the emission factors are underestimated since the experiments did not complete to dryness.

**5 CARB Architectural Coatings Survey Data**

Every four to five years, CARB conducts comprehensive surveys of architectural coatings sold in California to gather
information about the ingredients and sales with the goal of updating emission inventories. The response to the surveys
is mandatory and CARB ensures the validity of the data following extensive quality assurance and quality control
measures and the results accurately represent the sales volume in California. The data are publicly available (CARB,
2018). Speciation is based on reported product formulations and emissions data reflect applicable fate and transport
adjustments.
Figure 8a shows the trend in sales and emission estimates for the last five surveys from 1990 to 2014. The
sales volume increased significantly from 1990 to 2004, but was lower in 2014 as the industry was still recovering
from the economic recession in 2007-2009 that led to a sharp decline in construction spending (Bureau USC, 2020b).
During this time, coating emissions continuously decreased from 126 tons/day to < 27 tons/day in California. Most of
the sale volume was associated with water-borne coatings, and the fraction increased to ~ 75% of the total in 1990 to
~ 93% in 2013. The emissions, on the other hand, were dominated by solvent-borne coatings with > 72% of the total
in 1990 and half solvent-borne and water-borne each in 2014. This shows that the VOC content, and therefore the
emissions, of coatings in total have significantly decreased since the 1990s. Furthermore, the higher-emitting solvent
borne coatings are increasingly being replaced by water-borne products, which together resulted in this significant
emission reduction. The CARB survey also separates the sale and emissions into the different product categories (not
shown). Common water-borne paints, such as flat or low-gloss coatings and the accompanying primer, make up the
bulk of the total sales, but only about half of the emissions. In contrast, solvent-borne products such as stains,
varnishes, or rust preventative coatings are sold in lower volumes, but contribute the other half of the coating
emissions.
The South Coast Air Quality Management District (SCAQMD) also reports data from coatings sales and
emissions from 2008 to 2017 (reproduced in Fig. 8b). These data are part of the CARB survey and represent half of
the California-wide emissions. The sales in SCAQMD have been relatively steady ranging from 35-42 million gallons
with only slight increases since 2009, while the emissions have decreased significantly from 2008 to 2014 and were



constant around 11 tons/day from 2014 until 2017. This data set extends the CARB surveys and might indicate that
the overall emissions have not continued the steep decease after 2014, so that 2014 CARB data might still be
representative of the 2018 ambient measurements presented above.
The VOC ingredients of coatings reported in the CARB surveys have also changed significantly from 2005
to 2014. The left panels of Fig. 9a show the top 35 non-exempt VOC ingredients out of over 300 in the survey, together
with the exempt ingredients for 2005 and 2014 in the right panel. As was already clear from Fig. 8, the total amount
of VOC ingredients significantly decreased from 73 million pounds in 2005 to 23.6 million pounds in 2014, but also
the composition has changed. For example, in 2005 xylene was still a major ingredient of coatings but does not show
up in the top 35 ingredients in 2014. Even though the total amount of exempt ingredients stayed almost constant, the
relative amount has increased to almost 20% in 2014 from about 7% in 2005. The hydrocarbons from solvents are
generally the largest emissions, but small oxygenated VOCs, such as ethylene glycol, propylene glycol, acetone, and
ethanol, are also strongly emitted by paints and coatings and were detected as significant emitters in the laboratory
experiments. The ingredients measured as emissions in the ambient or laboratory experiments by the PTR-ToF are
indicated as solid bars in Fig. 9a for the 2014 data.
Most of the VOC ingredients shown in Fig. 9a are not unique to coatings products, but Texanol and PCBTF
are two compounds that are used only in coatings products as can be seen in Fig. 9b, where the fractions of VCP
emissions in the coatings category of various VOCs are shown. The compounds are sorted by their contribution to the
coatings category according to the FIVE-VCP inventory calculated using the method of McDonald et al. (2018a). The
only other compound besides Texanol and PCBTF that is predominantly attributed to coatings in the FIVE-VCP is
methylene chloride, but the amount used is too small to be a useful atmospheric tracer.
All the VOC ingredients collected by the CARB survey might not necessarily be emitted by the products; for
example Texanol airborne recoveries were between 25-90% depending on the paint and the substrate (Lin and Corsi,
2007) and in addition reactions and polymerization will occur in the production of the chemical products. The
ingredients that have been detected in either the evaporation experiments or the ambient measurements with the PTR-
ToF are indicated as solid bars for the 2014 data in Fig. 9a. Texanol emitted during evaporation of the primer/sealer
paint accounted for 13% of the measured VOC emissions. The ingredients reported in the CARB 2014 survey indicate
Texanol was 10% of the total VOC ingredient sales including exempt VOCs. The agreement between the laboratory
measurement of Texanol and the ingredients summary is reasonable considering the range in airborne recoveries.
PCBTF in the solvent-borne polyurethane only accounted for 0.2% of the total VOC mass as compared to the reported
ingredient sales contribution of 9%. These results demonstrate the challenges in generating a representative emissions
inventory from product sales as each product has a unique composition and ingredient sales do not necessarily equal
emissions. It is also possible that uses of PCBTF could differ between formulations available to consumers and those
used for professional industrial applications. Only a small selection of commercially available coatings were tested
here.
More details of the use of PCBTF and Texanol are shown in Fig. 10, where panel (b) shows that the use of
Texanol has strongly declined, while the use of PCBTF has increased such that they had comparable ingredients sales
in 2014. Both were around 10% of the total sale each. The pie charts in Fig. 10a show that PCBTF is mainly used as



a solvent in solvent-borne products, such as sealers, stains and polyurethane finishers. Texanol is used in water-borne products as a coalescent for latex and other paints (Lin and Corsi, 2007). In summary, the CARB survey results in Fig. 8, 9, and 10 indicate that PCBTF might be a good atmospheric tracer for solvent-borne coatings and Texanol for water-borne coatings, even though the use of Texanol is rapidly declining.

**6 Conclusion**

Mobile field measurements in urban areas show that compounds largely associated with architectural coatings, such as PCBTF and Texanol, were observed from point source locations near and around construction activity. Unlike other VCP emissions previously described by Gkatzelis et al. (2020a), these molecules do not correlate strongly with population density, which suggests that their emissions are not driven by wide-spread, individual usage. In contrast, the spatial and temporal patterns suggest that coating emissions are from discrete industrial applications, such as architectural and construction projects.

Headspace analysis measured with a PTR-ToF and GC front end confirmed the identity of many VOCs cataloged as ingredients in inventories compiled by CARB from architectural coating surveys. The "zero VOC" paint analyzed had low VOC emissions dominated by small oxygenates including methanol, ethanol, and acetone, with negligible emissions from smaller hydrocarbons and aromatics. The solvent-borne polyurethane stain emissions were compositionally different with the distribution shifted largely towards hydrocarbons and aromatics with a clear enhancement of PCBTF. The emission rates (mg s$^{-1}$) were calculated for each VOC and the total mass emitted as VOCs was calculated for the length of a controlled evaporation experiment and compared to total product mass emitted. The VOCs accounted for a range of 0.2-83% of the total mass emitted with the relative contribution mirroring the VOC content (g L$^{-1}$) reported by the manufacturer. Inorganic species such as ammonia were detected in water-borne coatings and likely contribute to some evaporative mass loss, though their contribution was not quantified in this study. The total VOC emission rates were highest within the first 6 minutes of application and for three of the four products, over 50% of the total VOC mass was emitted within the first two hours. These results highlight the importance of the initial evaporative emissions of coatings following application events, as they likely have important impacts on ozone.

Finally, reported sales and usage trends were compared to ingredients compiled in architectural coatings surveys, and show Texanol and PCBTF are unique to coatings. Although Texanol use has strongly declined and PCBTF has increased, they had comparable ingredients sales in 2014 at around 10% of the total. The prevalence and distinct usage of these VOCs support the assignment of PCBTF and Texanol as tracers for solvent- and water-borne coatings, respectively.

**Funding**: CES, MMC, GIG, JO, BCM, JP, KA JBG, MT, and CW acknowledge the CIRES Innovative Research Program and NOAA Cooperative Institute Agreement (NA17OAR4320101).

*Data availability*
The data from the laboratory tests are available on request. Ambient data from the NYICE 2018 are available here: https://esrl.noaa.gov/csl/groups/csl7/measurements/mobilelab/MobileLabNYICE/DataDownload/index.php?page=/csl/groups/csl7/measurements/mobilelab/MobileLabNYICE/DataDownload/





*Author Contributions*
MMC, CW, JBG, GIG, JO and carried out the laboratory experiments. JO designed and built the GC inlet system. Ambient measurements were collected by MMC, GIG, JBG, KA, and JP. FIVE-VCP inventory work was completed by BCM and MT. CES and CW conducted data analysis and wrote the manuscript. All authors contributed to the discussion and interpretation of the results.

*Competing Interests*
The authors declare that they have no conflict of interest.

*Disclaimer*
Mention of commercial products is for identification purposes only and does not imply endorsements.

*Acknowledgements*
The authors thank William Kuster for valuable conversations and assistance with the GC inlet system.



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



| Product | Mass emitted as measured VOCs | VOC Emission Factor | Labeled VOC content | Average VOC emission rate | Maximum VOC emission rate |
|---|---|---|---|---|---|
| | % | g kg$^{-1}$ | g L$^{-1}$ | mg day$^{-1}$ | mg hr$^{-1}$ |
| Polyurethane stain | 83 | 495 | 550 | 20.6 | 6.69 |
| Latex paint | 6.63 | 43.1 | 250 | 2.33 | 3.16 |
| Primer/sealer paint | 0.66 | 2.84 | 100 | 0.22 | 0.03 |
| "Zero VOC" paint | 0.17 | 0.71 | < 5 | 0.04 | 0.14 |

**Table 1: Summary of the evaporation experiments for four coating types including percentage of mass lost as measured**
**VOCs (%), VOC emission factors (g kg$^{-1}$ paint), reported product label VOC content (g L$^{-1}$), and the average and maximum**
**total VOC emission rates. Note: emission factors are lower-end estimates calculated based on the length of these experiments**

Figure 1: Summary of the PCBTF and Texanol measurements in New York City: (Map) Color-coded NOAA Mobile Laboratory drive track on a map of the population density. (a) Binned population density and mixing ratios. (b) Time series of CO, PCBTF, Texanol, and D5 siloxane for winter and summer months. The stagnation period during winter measurements is highlighted in orange. (c) The correlation plots of PCBTF and Texanol with CO for winter measurements. Slopes are calculated for the stagnation period at the end of the winter measurements (orange).



**Figure 2: (a)** Summary of the PCBTF and Texanol measurements from Chicago, New York City, Denver, and Pittsburgh. **(b)** Time series of PCBTF and D5 siloxane during Chicago drives.



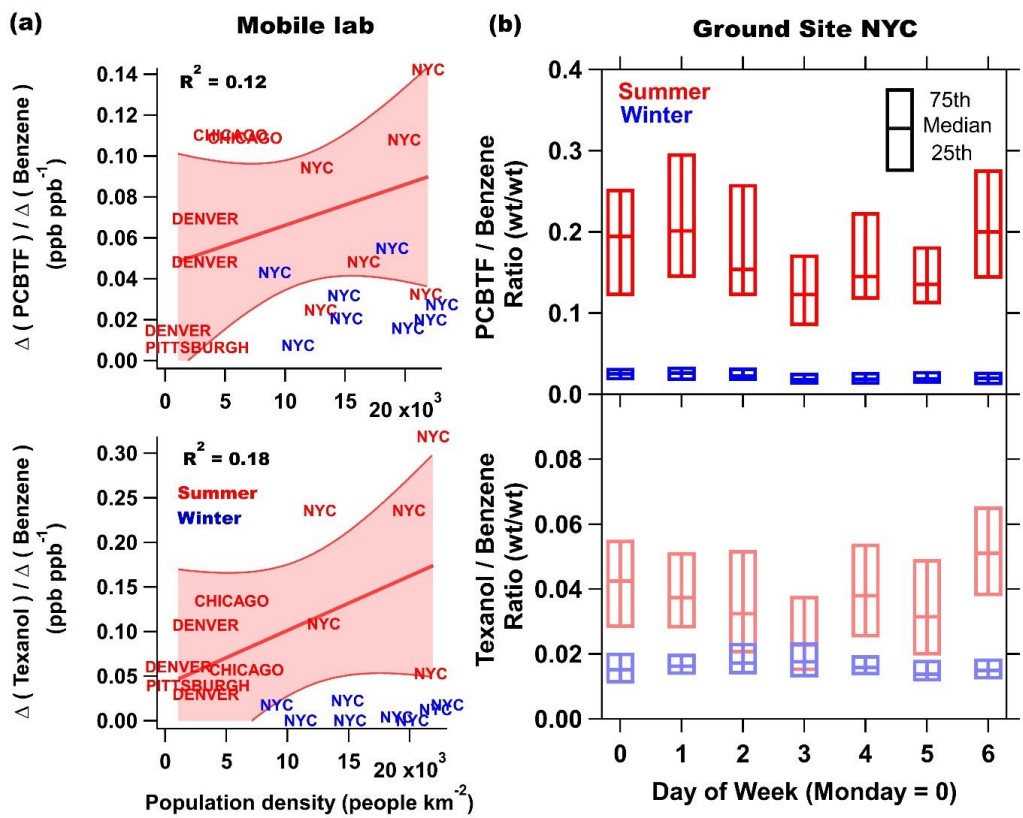

Figure 3: (a) The enhancement of PCBTF and Texanol relative to the enhancement of benzene versus the population density in summer (red) and winter (blue) in various cities. Denver and Chicago are represented by multiple drives. NYC is an average of all drives throughout the city, but separated by regions with high (19,000 - 23,000 people km$^{-2}$), medium (14,000-19,000 people km$^{-2}$), and low (9,000-14,000 people km$^{-2}$) population densities. (b) The summer and winter weekly profile of the PCBTF and Texanol versus benzene ratio.

**Figure 4: GC-PTR-ToF chromatogram from the headspace of a water-borne (wb), "zero VOC" low odor paint with ink and a solvent-borne (sb) polyurethane stain for (a) small oxygenates, (b) hydrocarbons, and (c) aromatics.**

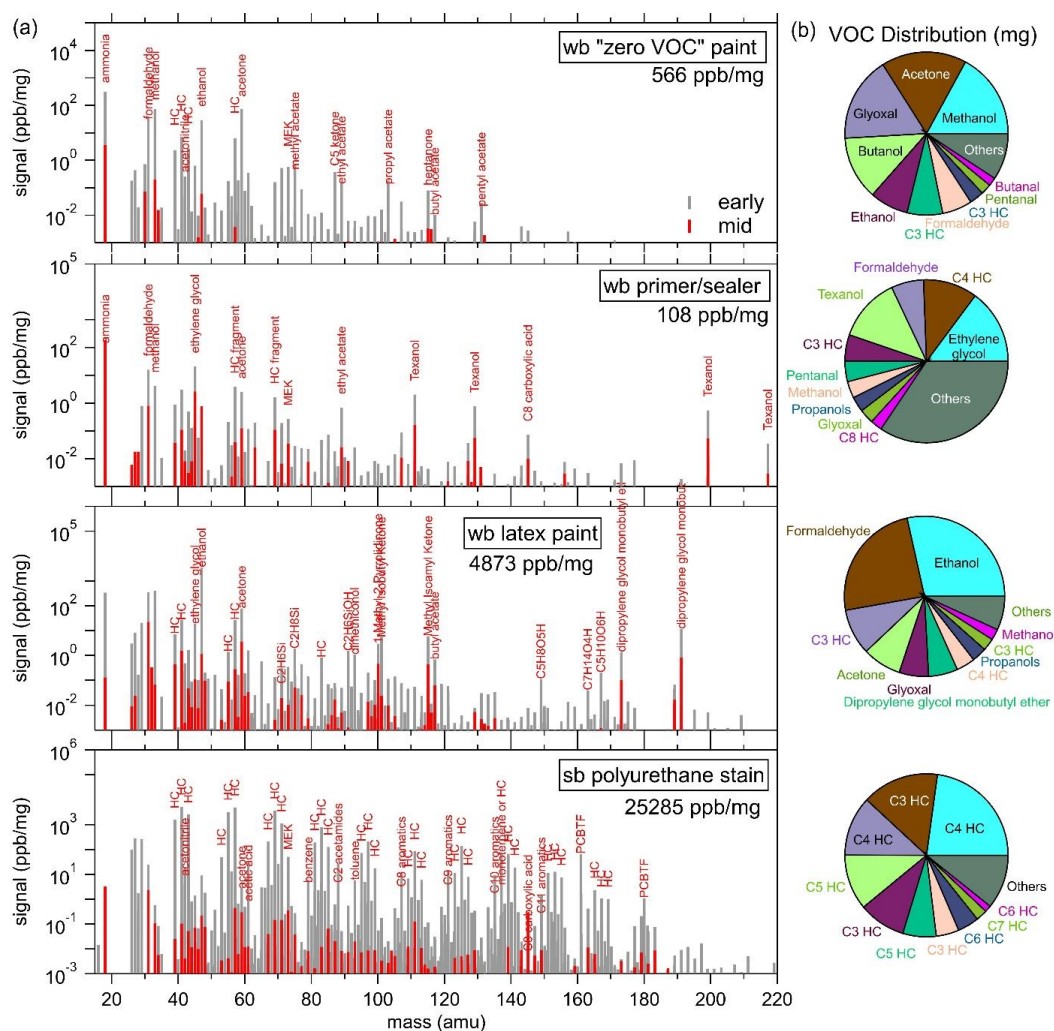

**Figure 5: (a) PTR-ToF-MS mass spectra of a water-borne (wb) low VOC paint, a wb primer/sealer paint, a wb latex paint, and a solvent-borne (sb) polyurethane stain, where the major peaks are labeled by their most likely identification. The values in ppbv are normalized to the weight loss of the product on the scale during the evaporation experiment. In each panel a mass spectra in the early part (grey) of the experiment and one in the middle part (red) are shown. The total VOC signal per mg of product is indicated in the legend. (b) Pie charts of the total mass emitted by individual VOCs during evaporation experiments.**





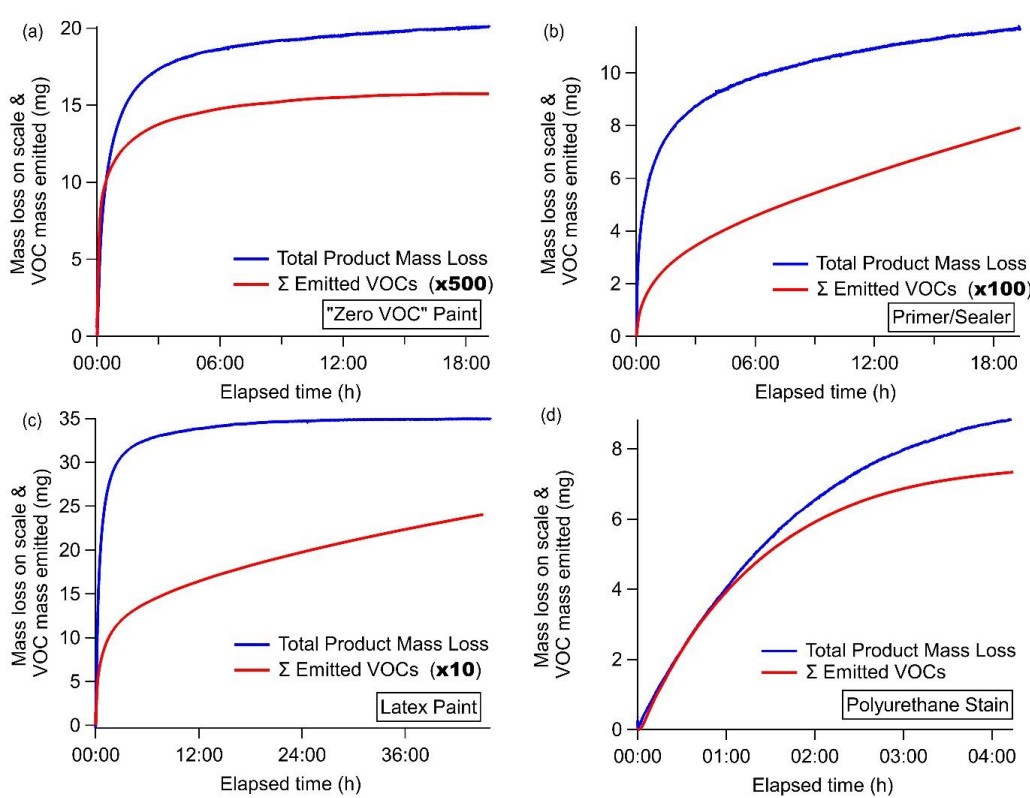

**Figure 6: Total product mass loss measured on the scale (blue) overlaid with the total mass emitted as VOCs (red) measured by the PTR-ToF-MS as a function of elapsed time. The emitted VOC scalar is indicated in the legends.**





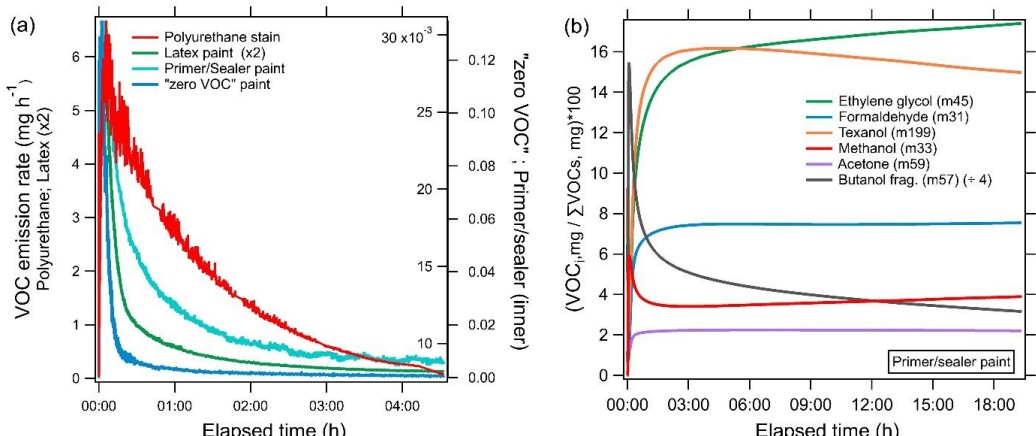

**Figure 7: (a) Emission rates (mg h⁻¹) of all measured VOCs summed during evaporation experiments of different coating types: polyurethane stain (red), latex paint (green), primer/sealer paint (cyan), "zero VOC" paint (blue). (b) Time series showing the percent of mass emitted as an individual VOC (VOC$_i$) to the total measured VOC mass for the evaporation of the sealer/primer paint. The most likely identity and detected mass of individual VOCs are indicated in the legend.**



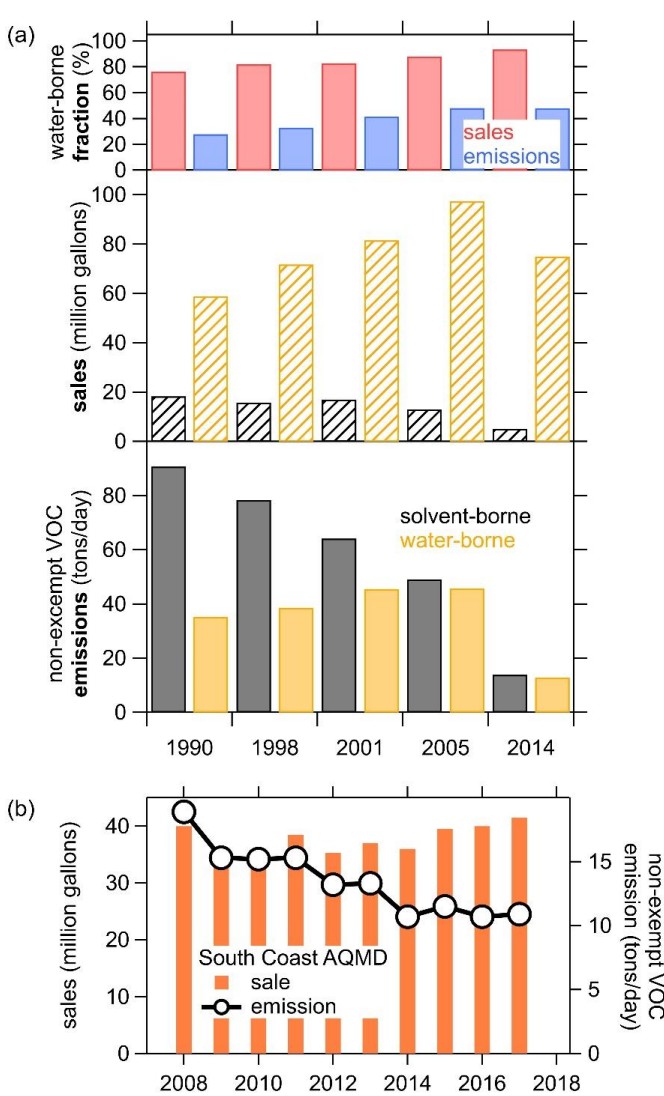

Figure 8: (a) Trends in paints and coatings sales and VOC emissions from the CARB coating surveys since 1990. (b) South Coast AQMD trend of coatings reproduced from: http://www.aqmd.gov/home/rules-compliance/compliance/vocs/architectural-coatings



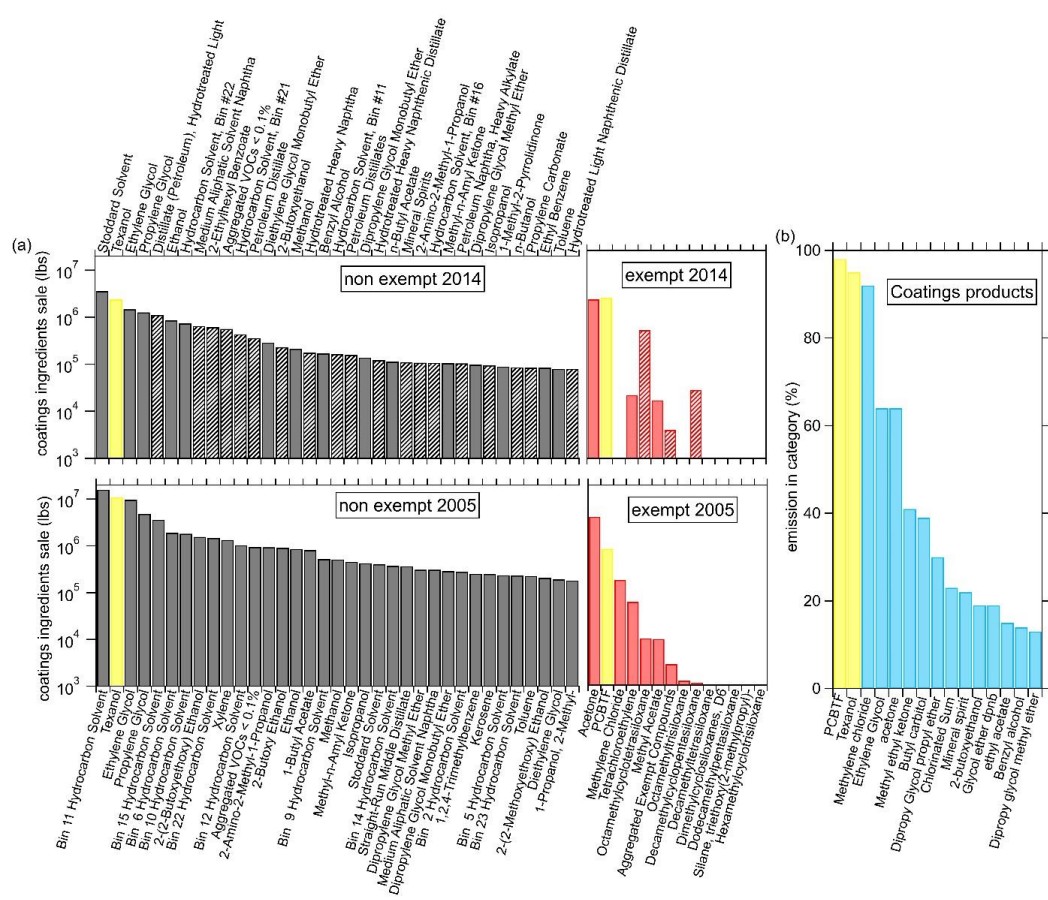

**Figure 9: (a) The sale of top 35 non-exempt and the exempt VOC ingredients for coatings in 2005 and 2014 in California together with the (b) fraction of emissions in the coatings category of the 15 highest VOCs as determined by the FIVE-VCP inventory calculated using the method of McDonald et al. (2018a). The solid bars in the 2014 data indicate compounds that were detected in ambient or product testing.**



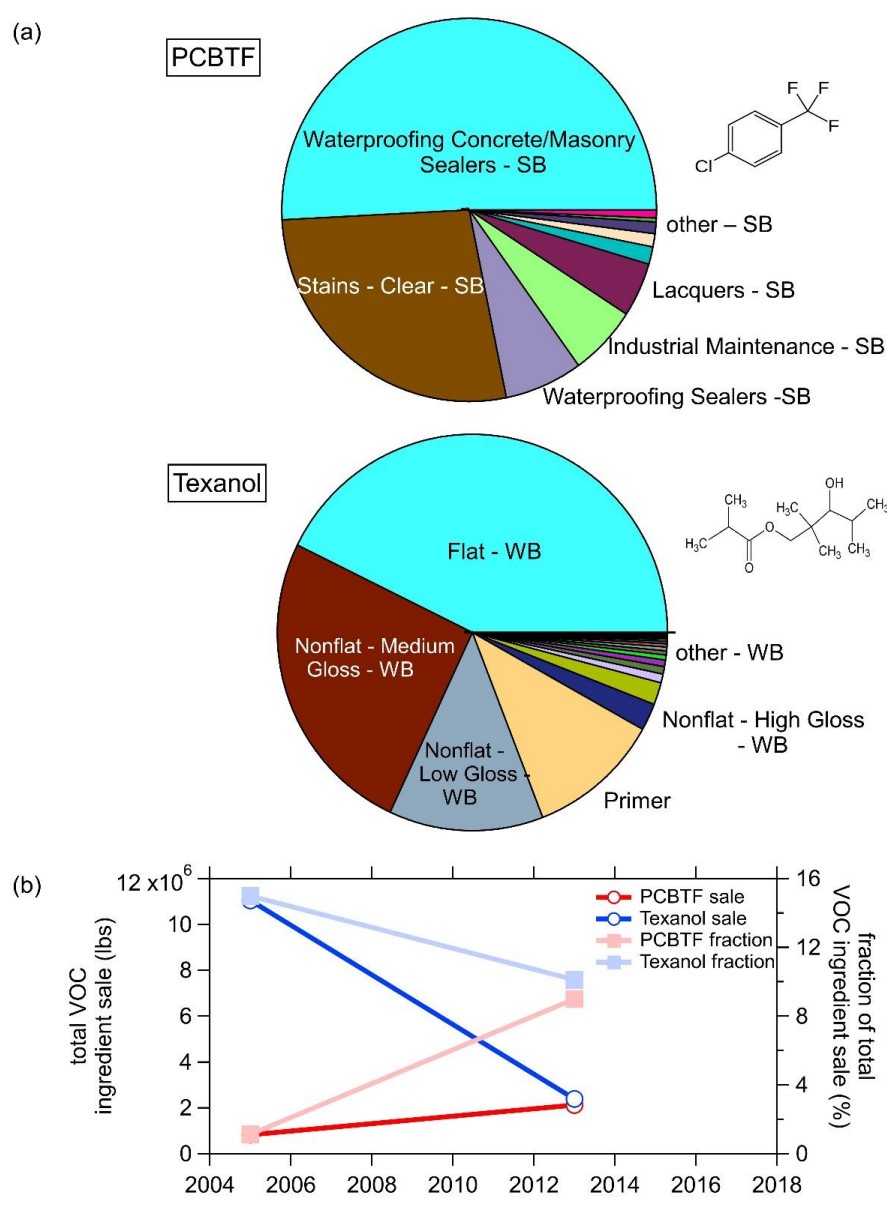

**Figure 10: (a) PCBTF and Texanol use in solvent-borne (SB) and water-borne (WB) product categories in California and their (b) trends in sales and emissions.**