# Peer review of "Volatile organic compound emissions from solvent- and water-borne coatings: compositional differences and tracer compound identifications"

_Atmospheric Chemistry and Physics, 2020_

## Referee Comment (RC1) · Anonymous Referee #1 · 8 Dec 2020

The authors present a compelling overview of Texanol and PCBTF emissions from coatings making the case for these compounds as specific tracers for water based and solvent based coatings. The authors accomplish this with an impressive synthesis of laboratory, ambient, and reported data. This paper makes a significant contribution to the study of VCPs, and I recommend it be published with minor revisions.

Mobile platform measurements show "hotspot" behavior for PCBTC, and perhaps Texanol, indicative of emission from manufacturing or construction sites. Neither compound correlates with transportation (CO) or PCP usage (Siloxane D5, population

density). Taking advantage of stagnant meteorological conditions, the authors also estimate emissions for the two proposed tracer compounds.

Furthermore, using ingredient and sale data for coatings (at least in California), they show these compounds are both commonly and specifically used in coatings. This establishes a good basis for a tracer: the compound is measurable and specific to the category.

Laboratory experiments are used to qualitatively examine the headspace emissions of 19 products. I believe that this information is useful, however I have some concerns about the experimental design. Are you concerned that the water trap could systematically remove water soluble compounds? How sensitive were recoveries for water soluble compounds to the water trap temperature? I understand that this is designed to be quantitative and not qualitative, but the fact that Texanol cannot be detected by this method suggests the data in Figure 4 should be interpreted with some caution. I am fine with the presentation of what was detected, but would like to see more of a disclaimer of what may not be detected.

Evaporation experiments show that among the category of "coatings", g VOC per kg product emissions can vary tremendously. It would be nice to have more than one example of a product within each subcategory (ie more than one "Latex paint"), however that may be outside the scope of this study. The authors show that the majority of VOC emissions occur shortly after application, and again the amount emitted and the time frame of emission vary within the category of coatings. The emissions presented in Figure 4 are great!

Before giving more specific notes, I would like encourage the authors to eventually add their PTR-ToF-MS findings to the PTR Library (Pagonis, D., Sekimoto, K. and de Gouw, J. A.: A library of proton-transfer reactions of H3O+ ions used for trace gas detection, J. Am. Soc. Mass Spectrom., 2019.) The application of GC before PTR should give rich fragmentation information for compounds that would help further research in this

field.

Some specific notes to follow:

Experimental methods: please list the exact masses on which you measure Texanol and PCBTC. Please list the temperature at which you conduct the laboratory experiments. One would suspect evaporation rates are highly temperature dependent. So much so that this may be worth mentioning in the discussion section as well!

Lines 33-35: VCPs as petrochemical organics: I would like a little clarification in the writing here. VCPs are defined as: VOCs that are ingredients of, and directly emitted from, coatings, adhesives, inks, personal care products, and cleaning agents (also pesticides if you would like to add). However, this sentence may leave the reader with the impression that they are necessarily petrochemical in origin, which is not true (Just as an example, McDonald 2018a include ethanol and monoterpenes as VCPs. I don't think either of these chemicals have a significant petrochemical source.) This sentence is technically correct, as many VCPs are of petrochemical origin, but I think it is especially important that while this term is somewhat young we really hammer down its strict definition.

Lines 42-43, ibid.

73 delete "a" 99 "was" to "were"

Line 212: "while emissions from VCPs scale linearly with the number of people" This is contradicted by the previous evidence showing that these two VCPs do not correlate strongly with population density. Do you mean PCPs rather than VCPs?

Lines 223-226 I see you are detecting Texanol on the dehydrated fragment, certainly expected for an alcohol. I would add that an isomer on this mass would be menthyl acetate, a terpenoid and flavor/odor additive. Seeing as you have calibrations of texanol, is there a fragmentation pattern that could be used to determine if there is interference in the ambient data?

Line 337, and figure 7a: should values these be normalized to the initial mass of the product used?

Figure 1b shows clearly shows PCBTF's "hotspot" behavior. Texanol, on the other hand does not show much use in the winter, which is helpful in showing that it does not follow CO or PCP patterns. However, in the summer the two largest spikes seem to align with large spikes of D5. Is this correct? If so, see my note regarding a possible interference from menthyl acetate.

Figure 2a: Legend is in units of people per square kilogram. Should be per square kilometer.

Figure 3: I think you could remove this figure. At the very least, remove the fits in 3a. The lines trick the eye into finding patterns that the R-squared says are not present.

Figure 5: Significant figures on the ppb/mg labels. 2? 3?

Figure 6: Consider using a left and right y axis rather than a VOC multiplier. I'm not certain, but it may read better.

Figure 10b: this figure is confusing to read. Consider another representation. Maybe a (grouped?) bar chart.

---

## Referee Comment (RC2) · Anonymous Referee #2 · 21 Dec 2020

This manuscript considers emissions of VOCs from volatile consumer products (VCPs). There is a specific focus on emissions from paints/coatings, and the use of two compounds (Texanol and PCBTF) as tracers. The manuscript is relevant to the recent interest in VCP emissions and atmospheric chemistry in the wake of declining emissions from combustion sources.

General comments: (1) The manuscript's results are broken into three main sections: (1) ambient mobile observations, (2) laboratory samples, and (3) analysis of an emissions inventory. These three sections are not connected very well, and came across

more as three stand-alone chunks rather than a single cohesive narrative. The inventory section in particular seemed to be the least connected to the other two main sections.

Comments on mobile sampling: (1) Figure 1 - what is the spatial resolution of the binning? It looks like the bins are strictly based on longitude. Is that correct? Does an east-west binning introduce any biases? I imagine that there are also strong north-south gradients in Manhattan.

(2) Are the data in Figure 1 from a single drive? Or are there road segments that were sampled on more than one occasion? If there are multiple passes over a road segment (or spatial bin), how were those handled?

(3) Line 183 notes isolated plumes - this suggests that emissions are episodic. If emissions are dominated by episodic events, it underscores the importance of how the mobile sampling data are binned and averaged. It's easy to imagine a case where the data end up being biased by capturing more plumes on one sampling day, or in one neighborhood, just by luck or chance. There are multiple papers that discuss spatial and temporal averaging needs for mobile sampling data (e.g., Messier et al, ES&T, 2018; E. Robinson et al, ES&T, 2019); the authors should at least acknowledge that this literature exists. If the data in Fig 1 are from a single driving pass, the authors should be clear about this, and they should also acknowledge that the data therefore represent a snapshot more than the typical or long-term spatial pattern.

(4) The correlations of PCBTF and Texanol with CO (Fig 1) are useful context. However, I would also like to see the correlation between PCBTF and Texanol. A visual inspection of the top part of Fig 1 suggests that these are not correlated very well. Given that the source(s) are the same, is this poor correlation expected? (It seems that this poor correlation is discussed in more detail in the section on the laboratory experiments. Nonetheless, it would be helpful to readers to give it some attention here.)

(5) Can the authors be more specific about what they mean by "industrial" uses of

paints and coatings? It seems like construction is one example of industrial use, but I can also imagine that industrial uses would include painting or coating of manufactured products. I think that these different types of industrial uses could be apparent in the data. E.g., are some of the Texanol spikes near large construction projects? In Figure 2 there seems to be a large area in northern Indiana with high PCBTF concentrations. Could this be emissions from non-construction industrial uses? My impression is that this region still has a lot of heavy industry.

(6) Figure 3 - Please define how the enhancement is determined (i.e., what is the background?).

Comments on laboratory tests: (1) Figure 4 and related discussion - PCBTF is only present in the solvent-borne paints, and not in the water-borne samples. However, prior to this point, the authors assert that PCBTF is a general paint/coating tracer. I think this needs to be clarified in the previous section.

(2) I do not see Texanol in Fig 4. Since that is the other tracer used in this work, I think it would be good to show it in this figure.

(3) One concern that I have about VCPs is that there can be so many sources of variability - e.g., between types of coatings (e.g., indoor vs outdoor, latex vs low VOC vs oil based), between specific formulations (e.g., different gloss levels of latex paint sold by the same manufacturer), and between manufacturers. Obviously it is impossible to capture all of these sources of variability in one paper, but they should at least be acknowledged. While the paint samples used here might be representative in the sense that they can be purchased commercially, there is a lot of potential variability that is not controlled for here.

(4) Figure 5 - are the total emission rates indicated in the figure (e.g., 586 ppb/mg for the zero VOC paint) the average over the whole experiment, or from a portion of the experiment?

(5) Figure 7b - does this show the instantaneous or integrated emissions? E.g., the purple line for acetone flattens out around 2% very quickly. Does this reflect that the volatile components are emitted quickly (including a big burst of acetone), or does acetone keep getting emitted at around 2% of total VOCs for the entire 19 hours of the experiment?

(6) Line 372-379 note that the trend in VOC emissions tracks the stated VOC content of the products. While that is true, there does not seem to be a linear relationship between stated VOC content and measured emissions. Latex paint emits 20x more VOCs than primer/sealer even though there is 2.5x the VOC content. There is a similarly large jump in VOC emissions between the stain and the latex paint for another factor of ~2 change in VOC content.

Comments on inventory section: (1) As I note above, I do not feel like this section is connected very well to the rest of the manuscript.

(2) This section focuses heavily on the emissions for California. How representative is CA of the rest of the US?

---

## Referee Comment (RC3) · Anonymous Referee #3 · 23 Dec 2020

VCPs' emissions have become a very important consideration in tropospheric chemistry studies, especially those focused on developed megacity environments. It is now essential to determine the major contributors to these emissions and specific tracer species that could assist in their isolation and quantification. From this perspective, this paper certainly would be useful for the scientific community. The authors have specifically focused on emissions from coatings-related products and have approached the source from multiple investigative dimensions including laboratory investigations, ambient measurements and exploration of existing chemical speciation surveys and

databases. The work is comprehensive in nature and attempts to encompass all manners of arguments to support the paper's objectives. Nevertheless, I have some concerns that need to be resolved before I could recommend the work for publication. They are listed below:

1. The introduction mostly focuses on VOCs which for the large part is fair for coatings-related products. Still, to present the complete picture, I believe some acknowledgement is warranted for small IVOCs (n-alkane equivalent volatility< C14) that may be present in solvents-based coating products. The authors do lend a quick word to LVP-VOCs that are currently exempt, but others have shown that LVP-VOCs are important for air quality and yet maybe underestimated due to instrumentation limitations and long emission timescales e.g. Khare and Gentner, 2018 ACP.

2. The experimental methods section could be better knitted. For example, as a reader, I'm confused about why I am reading 2.1.1 and 2.1.2 separately. As I understand it, 2.1.2 involves coupling a GC with PTR-ToF ahead of its inlet for improved separation that also permits detection of isomers. This could be easily merged with 2.1.1 and narrated cohesively.

3. Similarly, in laboratory measurements subsection (in methods), I would like to see some description (with perhaps a schematic in the SI) of the headspace sampling setup in the main text. Future experimental studies on VCPs could certainly benefit from it. The current description of laboratory methods left me with questions: Was the sample well-mixed (or re-mixed) within the container before headspace sampling occurred? Was the sample transferred to a new container for headspace sampling or was it done in the original product container? If the sample was transferred to a new container, was it collected from the core of the original product or scooped from the surface since the core may have more volatile content? What temperature was the product subjected to during emissions? How much time was allowed for air-sample equilibrium to be attained in the headspace? I understand that this was a qualitative investigation but the finer details are important to ascertain that the sample was not already partially

depleted of certain volatile constituents before sampling was conducted.

4. Lines 131-132: Please define extreme environmental conditions. Non-combustion emissions such as those from coatings would be temperature-dependent. Could excluding products used in high-temperature environments affect/bias your results in any way, especially the emission rates?

5. Lines 132-133: It is unclear how this was positioned. Was the PTR-ToF inlet exposed to an open product container? What was the distance between the inlet and the product? Was the direction of the airflow between the inlet and sample controlled? The issues go with point 3 above.

6. Lines 138-140,309: Please mention what was the range of the emissions velocity from the product's surface into the air flow. For quantification purposes, it is important that the emissions velocity is representative of environmental conditions in which the product is to be used. Otherwise, please provide an explanation for how the measured emission rates could be justified as relevant in real-world conditions, and also from a modeling perspective. What temperature was each sample subjected to during the chamber experiments?

7. It would probably help to keep the discussion sections consistent with the experimental subsections or vice versa. As a reader, I wasn't sure why I landed on a discussion section specifically focused on Texanol and PCBTF right after the methods. It might be better to present the bigger picture, broader details about the chemical speciation of tested products and field measurements, before narrowing it down to the proposed tracer compounds.

8. Section 4 "Headspace and Evaporation Measurements" could be written better. The paragraphs read disjointed and I had to scroll up and down repeatedly to properly understand the content. For example, line 244 marks initiation of the discussion of headspace sampling which in line 268 suddenly shifts to evaporation experiments without any ramp.

9. Lines 186-187: If PCBTF and Texanol are predominantly from construction activities, some explanation is required for why their mixing ratios were dominated by short, isolated plumes? I would expect construction activity in an area to be decaying but continuous source of the tracer, especially if the source is industrial coatings. Or is this because the mobile lab was in continuous movement and saw a temporary spike when it happened to be passing by a source? I think it is the latter based on the opening statements in the conclusion paragraph. However, it is important to clarify this, or else source behavior could be easily misunderstood.

10. Lines 349-351: The authors mention that acetone and formaldehyde did not emit as expected based on thermodynamic considerations but leave it at that calling it "complicated". I found the trends in Figure 7b very interesting and would very much like to see some explanation, even if just reasonable speculation from the authors for these observed trends.

11. Figure 2b could go into the SI. Since these are mobile measurements, the concentration measured is not singularly dependent on time of day but also the location of the vehicle. Hence, the figure is confusing and not substantial enough in my perspective to warrant a spot in the main text. Would the highest peaks still be observed between 2-3 PM if the vehicle was at some other location? The figure caption should clearly mention that these are mobile laboratory-based ambient measurements.

Minor points:

-Line 244-245: Nitpicking here but measuring the headspace by PTR-ToF is unclear. Sampled vapors from the headspace is more accurate.

-Fig. vs. Figure is used inconsistently in the text. Line 268-"Figure 5a", line 277-"Fig. 5b". Similarly for 6. Line 310 says "Figure 6", line 314- "Fig. 6". Please check others.

-Line 392: increased "from".

-Figure 3: (a) has illegible overlap in the figure.

-Figure 5: is clumsy. Compound identifications overlap and are not readable in several places. Consider using arrows where necessary. Some compound names are not fully printed in the spectra.

-All across the figures, the legends sometimes start with block letters and other times with small letters. Please correct the inconsistency.

---

## Author Response (AR1)

**Response to Referees**

We thank all three referees for their helpful reviews and constructive commentary. The manuscript has been revised accordingly with our responses interspersed in black text.

Anonymous Referee #1

The authors present a compelling overview of Texanol and PCBTF emissions from coatings making the case for these compounds as specific tracers for water based and solvent based coatings. The authors accomplish this with an impressive synthesis of laboratory, ambient, and reported data. This paper makes a significant contribution to the study of VCPs, and I recommend it be published with minor revisions.

Mobile platform measurements show "hotspot" behavior for PCBTC, and perhaps Texanol, indicative of emission from manufacturing or construction sites. Neither compound correlates with transportation (CO) or PCP usage (Siloxane D5, population density). Taking advantage of stagnant meteorological conditions, the authors also estimate emissions for the two proposed tracer compounds.

Furthermore, using ingredient and sale data for coatings (at least in California), they show these compounds are both commonly and specifically used in coatings. This establishes a good basis for a tracer: the compound is measurable and specific to the category.

Laboratory experiments are used to qualitatively examine the headspace emissions of 19 products. I believe that this information is useful, however I have some concerns about the experimental design. Are you concerned that the water trap could systematically remove water soluble compounds? How sensitive were recoveries for water soluble compounds to the water trap temperature? I understand that this is designed to be quantitative and not qualitative, but the fact that Texanol cannot be detected by this method suggests the data in Figure 4 should be interpreted with some caution. I am fine with the presentation of what was detected, but would like to see more of a disclaimer of what may not be detected.

We agree that the sample pre-concentration and GC system was not thoroughly characterized to analyze all possible volatile and semi-volatile compounds. We specifically designed the system to capture the majority of compounds in the C3-C10 range that are typically observed in ambient air, while simultaneously maintaining a relatively short cycle time. With these objectives, the PTR-ToF-MS is able to sample in-situ for 75% of the time, while periodically taking GC samples to obtain additional information about structural isomers and to pre-concentrate samples to lower the detection limits of certain compounds.

The following text was added to section 4 (Headspace and Evaporation Measurements):

"As a stand-alone instrument, the PTR-ToF-MS has the distinct advantage of having a rapid sampling time (~0.1-5 seconds). The GC-inlet system requires ~10-15 minutes for the sequence of sample preconcentration, injection, GC separation, and detection. It was originally developed to collect a sample in the GC during sampling with the PTR-ToF-MS followed by interrupting normal sampling to analyze the chromatogram and take advantage of chromatographic separation. This setup still maintains >75% of the time dedicated to in-situ PTR-ToF sampling. The GC system is capable of detecting C3-C10 compounds and development included calibration standards in that range, which keeps the sampling and analysis times relatively short while capturing the majority of typical VOCs observed in ambient air. Further characterization of larger and less volatile compounds such as Texanol would have required a different GC system with significantly longer cycle times."

The following text was added to the supplement:

"Calibration standards containing components with carbon numbers ranging from acetone (C3) to monoterpenes (C10) were tested to ensure that compounds were trapped and released through the water trap, sample trap and sampling lines and could be quantitatively detected by the PTR-ToF-MS. Both polar and non-polar compounds were tested including acetone, isoprene, BTEX compounds, indene,

crotonaldehyde, 2,3, methyl buten-2ol, and methyl ethyl ketone. To analyze larger (and less volatile) molecules such as C12-C15 alkenes, sesquiterpenes, tri-isopropyl benzene or Texanol (C12), the trapping system would have required modifications. Among these are higher transfer line temperatures, higher water trap temperature, longer sampling times, higher GC oven temperature and possibly a different analytical column. Since the initial characterization included identification and quantification of the vast majority of VOCs observed in ambient air with a total cycle time of approximately 15 minutes, the design criteria were met. Further efforts to analyze heavier or lighter compounds (e.g. acetylene) were not pursued."

Evaporation experiments show that among the category of "coatings", g VOC per kg product emissions can vary tremendously. It would be nice to have more than one example of a product within each subcategory (ie more than one "Latex paint"), however that may be outside the scope of this study. The authors show that the majority of VOC emissions occur shortly after application, and again the amount emitted and the time frame of emission vary within the category of coatings. The emissions presented in Figure 4 are great!

As described in Section 2 (Laboratory Measurements), the headspace of nineteen total products were tested, but only a few were sub-selected for evaporation experiments. The nineteen products varied tremendously and we focused on selecting products for the evaporation experiments that ranged in total VOC-content, but also included both water-borne and solvent-borne products that emitted frequently detected compounds, such as Texanol and PCBTF.

Before giving more specific notes, I would like encourage the authors to eventually add their PTR-ToF-MS findings to the PTR Library (Pagonis, D., Sekimoto, K. and de Gouw, J. A.: A library of proton-transfer reactions of H3O+ ions used for trace gas detection, J. Am. Soc. Mass Spectrom., 2019.) The application of GC before PTR should give rich fragmentation information for compounds that would help further research in this field.

We thank the referee for their suggestion and will continue to be in touch with Joost de Gouw to add to the PTR-library spreadsheet as additional VOCs are identified and fragmentation patterns are measured with the GC-front end.

Some specific notes to follow:

Experimental methods: please list the exact masses on which you measure Texanol and PCBTC.

Thank you for pointing out this was not included in the text. This information is included in Gkatzelis et al. (2021a), however, it should also be included in this manuscript. We have added the following text to the Experimental Methods Instrument subsection: "Texanol ($C_{12}H_{24}O_3$) was measured as a dehydrated fragment at $m/z$ 199.169 ($C_{12}H_{22}O_2 \cdot H^+$) and PCBTF ($C_7H_4ClF_3$) was detected at $m/z$ 160.996 ($C_7H_4ClF_2$) from the loss of fluorine."

Please list the temperature at which you conduct the laboratory experiments. One would suspect evaporation rates are highly temperature dependent. So much so that this may be worth mentioning in the discussion section as well!

We have added in the Laboratory Measurement section that the evaporation experiments were performed at room temperature. At the suggestion of Referee #3, we have added additional experimental details about the headspace analyses in the same section.

Lines 33-35: VCPs as petrochemical organics: I would like a little clarification in the writing here. VCPs are defined as: VOCs that are ingredients of, and directly emitted from, coatings, adhesives, inks, personal care products, and cleaning agents (also pesticides if you would like to add). However, this sentence may leave the reader with the impression that they are necessarily petrochemical in origin, which is not true (Just as an example, McDonald 2018a include ethanol and monoterpenes as VCPs. I don't think either of

these chemicals have a significant petrochemical source.) This sentence is technically correct, as many VCPs are of petrochemical origin, but I think it is especially important that while this term is somewhat young we really hammer down its strict definition.

We thank the reviewer for highlighting this detail and agree that while VCPs is a relatively infant term, we should limit confusion where possible. We have modified the following text to emphasize these are VOC emissions from volatile chemical products, removed the mention of "photochemical organics," and added "pesticides" to the list as suggested.

Adjusted text: "After decades of declining mixing ratios of volatile organic compounds (VOCs) in urban areas from combustion-related processes (Warneke et al., 2012), emissions from volatile chemical products (VCPs = coatings, adhesives, inks, personal care products, pesticides, and cleaning agents) have emerged as a major source of VOCs in the urban atmosphere (McDonald et al., 2018a)."

Lines 42-43, ibid.

We believe that the emphasis of the following sentences is important as it highlights the relative importance of both personal care products and coatings and therefore prefer to keep the text as is.

L73 delete "a" : Corrected L99 "was" to "were" Corrected

Line 212: "while emissions from VCPs scale linearly with the number of people" This is contradicted by the previous evidence showing that these two VCPs do not correlate strongly with population density. Do you mean PCPs rather than VCPs?

We thank the reviewer for their attention to detail and we will emphasize that personal care products and other daily use VCPs have been shown to scale linearly with population density to avoid later confusion when Texanol and PCBTF are shown to have poor correlations with population density.

Text at L212 changed to: "while emissions from personal care products and other daily-use VCPs driven by human activity scale linearly with the number of people"

Lines 223-226 I see you are detecting Texanol on the dehydrated fragment, certainly expected for an alcohol. I would add that an isomer on this mass would be menthyl acetate, a terpenoid and flavor/odor additive. Seeing as you have calibrations of texanol, is there a fragmentation pattern that could be used to determine if there is interference in the ambient data?

Gkatzelis et al. (2021a) noted that the signal at $m/z$ 199 in the winter was expected to be predominantly related to coatings, however, they did note this mass might be influenced by other compounds in the summer, especially those formed via chemical processing based on persistent high levels during midday hours and positive-matrix-factorization (PMF) results of the same dataset (Gkatzelis et al., 2021b). We have made note of these summer-time ambient interferences later in this same section (Ambient Measurements of PCBTF and Texanol).

Line 337, and figure 7a: should values these be normalized to the initial mass of the product used?

We define the emission rate here as the mass emitted per unit time. The mass concentration of each species (mg m$^{-3}$) was multiplied by the gas flow rate across the product through the Teflon chamber. The total mass emitted is normalized by the initial mass of the product when calculating an emission factor in units of mass emitted per mass applied (g kg$^{-1}$).

Figure 1b shows clearly shows PCBTF's "hotspot" behavior. Texanol, on the other hand does not show much use in the winter, which is helpful in showing that it does not follow CO or PCP patterns. However, in the summer the two largest spikes seem to align with large spikes of D5. Is this correct? If so, see my note regarding a possible interference from menthyl acetate.

The actual correlation between D5 and Texanol during the entire summer sampling period is poor ($R^2 \sim 0.2$). Occasionally, the enhancements are well correlated over short periods of time. This might be

related to oxidation products detected at *m/z* during the summer or other possible interferences as you suggest. This is described in detail in Section 3 (Ambient Measurements). Figure 1a also emphasizes the "hotspot" behavior of Texanol as you see the largest enhancement in the northeast region of the map where population density is low.

Figure 2a: Legend is in units of people per square kilogram. Should be per square kilometer. Corrected

Figure 3: I think you could remove this figure. At the very least, remove the fits in 3a. The lines trick the eye into finding patterns that the R-squared says are not present.

We believe Figure 3 is a useful as its correlation with population is shown to be weak and can be compared to other VCP tracers in Gkatzelis et al. (2021a). Figure 2 shows that the highest mixing ratios generally occur in areas of the highest population density within a city, however, enhancements are dominated by short, spatially isolated plumes. Figure 3, ties all the sampling cities together that varied in absolute population density. Additionally, we see a strong separation between summer and winter measurements for both potential tracers.

Figure 5: Significant figures on the ppb/mg labels. 2? 3? Corrected

Figure 6: Consider using a left and right y axis rather than a VOC multiplier. I'm not certain, but it may read better.

The first graphs generated for these plots used two axes, however, we believe using the same axis with the multiplier bolded better illustrated the mass and time-dependent overlap between the two methods.

Figure 10b: this figure is confusing to read. Consider another representation. Maybe a (grouped?) bar chart.

We believe it is important to show that the relative fraction of PCBTF and Texanol use is comparable even with significant changes in total sales in recent years. We generated a figure using a grouped bar chart as suggested by the Referee and found similar issues as the data currently available remains limited. As such we believe the current Figure still effectively demonstrates that PCBTF and Texanol use is comparable even with declining Texanol sales.

Referee #2

This manuscript considers emissions of VOCs from volatile consumer products (VCPs). There is a specific focus on emissions from paints/coatings, and the use of two compounds (Texanol and PCBTF) as tracers. The manuscript is relevant to the recent interest in VCP emissions and atmospheric chemistry in the wake of declining emissions from combustion sources.

General comments: (1) The manuscript's results are broken into three main sections: (1) ambient mobile observations, (2) laboratory samples, and (3) analysis of an emissions inventory. These three sections are not connected very well, and came across more as three stand-alone chunks rather than a single cohesive narrative. The inventory section in particular seemed to be the least connected to the other two main sections.

The focus of this study is to build on the work by Gkatzelis et al. (2021a) to help unambiguously identify VOC tracer compounds linked to coatings observed in ambient measurements. In order to better tie the three sections together, we've added the following text to the introduction:

"Presented here are ambient and laboratory measurements of vapors emitted from coatings using proton-transfer-reaction time-of-flight mass spectrometry (PTR-ToF-MS) to evaluate potential tracer compounds. PCBTF and Texanol will be shown to be detected at ambient levels, are VOCs emitted primarily by coatings products, and are unique VOCs prevalent in emissions inventories"

The remaining text in the introduction emphasizes the order by which we evaluate the performance of these VOCs as coatings tracers: (1) ambient measurements, (2) laboratory measurements, and (3)

inventory evaluations. Hopefully, this description in the introduction better emphasizes the need for all three sections and we believe the information is presented in the correct order.

Comments on mobile sampling: (1) Figure 1 - what is the spatial resolution of the binning? It looks like the bins are strictly based on longitude. Is that correct? Does an east-west binning introduce any biases? I imagine that there are also strong northsouth gradients in Manhattan. (2) Are the data in Figure 1 from a single drive? Or are there road segments that were sampled on more than one occasion? If there are multiple passes over a road segment (or spatial bin), how were those handled? (3) Line 183 notes isolated plumes - this suggests that emissions are episodic. If emissions are dominated by episodic events, it underscores the importance of how the mobile sampling data are binned and averaged. It's easy to imagine a case where the data end up being biased by capturing more plumes on one sampling day, or in one neighborhood, just by luck or chance. There are multiple papers that discuss spatial and temporal averaging needs for mobile sampling data (e.g., Messier et al, ES&T, 2018; E. Robinson et al, ES&T, 2019); the authors should at least acknowledge that this literature exists. If the data in Fig 1 are from a single driving pass, the authors should be clear about this, and they should also acknowledge that the data therefore represent a snapshot more than the typical or long-term spatial pattern.

We'll address the first three questions concerning mobile emissions together as there is significant overlap. The mobile laboratory measurements are shown as bars with median concentrations binned by longitude at 0.02 degrees. The median value reduces the effect of outlier observations on the comparison. Drives were conducted for several days in both the winter and summer. There are instances of repeated visits across different sampling days or within the same longitudinal bins, and in these instances the average was taken. We will add text to detail this in the manuscript. In NYC, the population density varies across its five boroughs and the largest population gradient is seen longitudinally. Ground-site measurements were used to investigate longer-term temporal patterns, while the aim of mobile sampling was to show spatial patterns. Coggon et al. (2018) and Gkatzelis et al. (2021a) have shown select volatile chemical products used on a daily-basis, such as personal care products (i.e. D5-siloxane), have strong correlations with population density. Gkatzelis et al. (2021a) shows these strong correlations for other VCP tracers such as monoterpenes (fragrances) and D4-siloxane (adhesives). The aim of this section is to show that the correlation with population density is not as strong for PCBTF and Texanol. As the Referee mentioned the emissions for these two compounds were driven by episodic events and this conclusion is supported by its poor correlation with population density while other VCPs continue to show a strong dependence in several cities. We make it clear in the manuscript that these are "short, spatially isolated plumes" and the ground-site measurements represent more-long term temporal patterns. We add a sentence to clarify the intention of the mobile measurements as the aim was not to identify long-term spatial patterns.

Added details of sampling strategy: "Multiple drives across the two seasons were conducted and longitudinally overlapping drives were averaged."

"The mobile sampling strategy relied on either no or a relatively small number of repeat visits in both seasons and therefore does not represent long-term spatial patterns (Messier et al., 2018; Robinson et al., 2019), however, we might expect emissions from coatings to not show strong long-term spatial patterns as construction activities are generally not permanently located."

Added to the references: Messier, K. P., Chambliss, S. E., Gani, S., Alvarez, R., Brauer M., Choi, J. J., Hamburg, S. P., Kerckhoffs, J., LaFranchi, B., Lunden, M. M., Marshall, J. D., Portier, C. J., Roy, A., Szpiro, A. A., Vermeulen, R. C. H., Apte, J. S.: Mapping air pollution with Google Street View cars: Efficient approaches with mobile monitoring and land use regression, Environ. Sci. &Technol., 52, 12563−12572, doi: 10.1021/acs.est.8b03395, 2018.

Added to the references: Robinson, E. S., Shah, R. U., Messier, K., Gu, P., Li, H. Z., Apte, J. S., Robinson, A. L., Presto, A. A.: Land-Use regression modeling of source-resolved fine particulate matter components from mobile sampling, Environ. Sci. & Technol., 53, 8925−8937, doi:10.1021/acs.est.9b01897, 2019.

(4) The correlations of PCBTF and Texanol with CO (Fig 1) are useful context. However, I would also like to see the correlation between PCBTF and Texanol. A visual inspection of the top part of Fig 1 suggests that these are not correlated very well. Given that the source(s) are the same, is this poor correlation expected? (It seems that this poor correlation is discussed in more detail in the section on the laboratory experiments. Nonetheless, it would be helpful to readers to give it some attention here.)

Your visual inspection is correct, the correlation between PCBTF and Texanol is not strong during the summer season ($R^2<0.1$). This poor correlation is not unexpected since these compounds are emitted by different types of coatings (e.g., water-borne or solvent-borne) and are often used for different applications. We add the following sentence to this section to make this clear.

Added text: "Enhancements of PCBTF do not necessarily coincide with Texanol enhancements. The poor correlation between PCBTF and Texanol ($R^2 < 0.1$) is not unexpected since the primary source varies by product formulation and usage (e.g., water- or solvent-borne)."

(5) Can the authors be more specific about what they mean by "industrial" uses of paints and coatings? It seems like construction is one example of industrial use, but I can also imagine that industrial uses would include painting or coating of manufactured products. I think that these different types of industrial uses could be apparent in the data. E.g., are some of the Texanol spikes near large construction projects? In Figure 2 there seems to be a large area in northern Indiana with high PCBTF concentrations. Could this be emissions from non-construction industrial uses? My impression is that this region still has a lot of heavy industry.

We thank the reviewer for pointing out the confusion, as we must be clearer in our separation of architectural and industrial coatings as they are regulated separately. The definition of architectural coatings are products that are applied to stationary structures and their accessories. They include house paints, stains, industrial maintenance coatings, traffic coatings, and many other products. Industrial maintenance coatings formulated for their chemical and corrosion resistance are a subcategory of architectural coatings when applied to stationary structures. Generally, coatings applied in shop applications or to non-stationary structures are not considered architectural coatings and instead are categorized as industrial coatings. The formulations of many architectural and industrial coatings can be similar, therefore, when using Texanol or PCBTF as tracers, we cannot distinguish between the sources of architectural or industrial coatings. We've added the following text to the introduction to emphasize the distinction between these two categories:

"Architectural coatings are defined as products applied to stationary structures and their accessories, whereas coatings applied in shop applications or to non-stationary structures are categorized as industrial coatings. Both coating types can be utilized in industrial applications (e.g., construction or manufacturing activities)."

In the manuscript a lot of the confusion lies in that we had generalized the term "industrial" to include the activities of applying coatings to stationary objects. As an example, we included the industry of construction into the phrase "industrial usage patterns." You are correct that there could be non-construction industrial uses in the manufacturing sector. To clarify we've changed the text to state: "emissions from coatings are influenced by industrial usage patterns, such as construction or manufacturing activity" and have eliminated the use of "industrial" in other areas of the manuscript to limit confusion.

With regards to your suggestion that different types of industrial uses could be apparent in the data, manufacturing processes generally occur indoors, often with VOC capturing systems. The indoor/outdoor exchange can still impact urban air, though it is likely strong enhancements would be only observed at or near manufacturing sites. The enhancements in PCBTF in the area of northern Indiana and in other areas of the U.S. were qualitatively observed near construction sites or near asphalting road activities. It would be a useful goal for future studies to detail enhancements near manufacturing sites, though this is outside the scope of this current manuscript.

(6) Figure 3 - Please define how the enhancement is determined (i.e., what is the background?).

We report enhancement ratios relative to benzene. Enhancement ratios are calculated using median concentrations, which reduces the impact of local emissions on calculated ratios (that would also impact the slope of a scatter plot). We subtract the background median using measurements upwind of the cities that exhibited the lowest mixing ratios from the median using measurements from regions where the population density was the highest.

We've added the following text to the Figure 3 description:

"Enhancements were calculated by subtracting the background median taken from measurements upwind of cities with the lowest mixing ratios from median concentrations from regions where population density was the highest."

Comments on laboratory tests: (1) Figure 4 and related discussion - PCBTF is only present in the solvent-borne paints, and not in the water-borne samples. However, prior to this point, the authors assert that PCBTF is a general paint/coating tracer. I think this needs to be clarified in the previous section.

Based on the Referee's earlier comments we've added to the ambient measurement section a discussion of the correlation between PCBTF and Texanol and how the poor correlation is not unexpected because the primary sources vary depending on whether it is a solvent- or water-borne formulation. Additionally, we now modify the last sentence of the introduction to state: "Coating surveys generally agree with the ambient and laboratory measurements and support the assignment of Texanol and PCBTF as atmospheric tracers for water-borne and solvent-borne coatings, respectively." This now alludes to a separation between water- and solvent-borne tracers at several locations in the manuscript before the laboratory measurement section starts.

(2) I do not see Texanol in Fig 4. Since that is the other tracer used in this work, I think it would be good to show it in this figure.

We detail in Section 4 that the mass for Texanol and its fragments were observed in the direct PTR-ToF sampling of the headspace of several water-borne products, but pure Texanol injections into the GC-PTR-ToF-MS demonstrated that Texanol does not elute from the GC column either because it is lost to the water trap or has too low of a vapor pressure to elute from the column under the selected temperature program. Based on suggestions by Referee #1 we've added additional text in the main manuscript and supplement detailing the characterization of the GC-system.

(3) One concern that I have about VCPs is that there can be so many sources of variability - e.g., between types of coatings (e.g., indoor vs outdoor, latex vs low VOC vs oil based), between specific formulations (e.g., different gloss levels of latex paint sold by the same manufacturer), and between manufacturers. Obviously it is impossible to capture all of these sources of variability in one paper, but they should at least be acknowledged. While the paint samples used here might be representative in the sense that they can be purchased commercially, there is a lot of potential variability that is not controlled for here.

We agree that coatings formulas vary considerably and we try to illustrate this by listing all the types of coatings in the introduction and also highlighted we only sampled commercially available products and industrial maintenance coatings formulated to withstand harsher environmental conditions were not

included. In order to acknowledge the limitations of this study, we add the following disclaimer into the Experimental Methods section:

Added text: "The variability in product formulation and usage is significant and the coatings in this study were selected based on availability, total VOC content, and similarity between other products tested, therefore the results do not fully capture the emissions variability of all coatings"

(4) Figure 5 - are the total emission rates indicated in the figure (e.g., 586 ppb/mg for the zero VOC paint) the average over the whole experiment, or from a portion of the experiment?

Line 275 mentions the sums of all the measured PTR-TOF signals in the "early" mass scan is identified by the labels. In order to make this clearer, we also add to the figure caption "taken from the early mass scan"

(5) Figure 7b - does this show the instantaneous or integrated emissions? E.g., the purple line for acetone flattens out around 2% very quickly. Does this reflect that the volatile components are emitted quickly (including a big burst of acetone), or does acetone keep getting emitted at around 2% of total VOCs for the entire 19 hours of the experiment?

Figure 7b shows the relative contribution of individual species to the total integrated mass over time. Therefore, the largest mass of acetone is emitted early and its relative contribution to the total mass emitted gradually levels off. After a few hours the species contributing to total VOC mass is largely driven by only a few compounds (e.g. ethylene glycol). To make this clearer the following text was added to the figure caption.

 "(b) Time series showing the percent of integrated mass emitted as an individual VOC ($VOC_i$) to the total measured integrated VOC mass for the evaporation of the sealer/primer paint."

(6) Line 372-379 note that the trend in VOC emissions tracks the stated VOC content of the products. While that is true, there does not seem to be a linear relationship between stated VOC content and measured emissions. Latex paint emits 20x more VOCs than primer/sealer even though there is 2.5x the VOC content. There is a similarly large jump in VOC emissions between the stain and the latex paint for another factor of ~2 change in VOC content.

We thank the referee for their attention to detail and agree that the percent emitted and content reported in g VOC/L cannot be compared directly as the density of each product would need to be considered and known to calculate a mass emitted that is directly comparable. The goal was to emphasize the qualitative agreement between the manufacturer's content label and we have modified the wording to this effect.

Added text: "The actual density of each product would need to be known to calculate the VOC mass emitted from the content labels, thus the comparisons described above are qualitative."

Comments on inventory section: (1) As I note above, I do not feel like this section is connected very well to the rest of the manuscript. (2) This section focuses heavily on the emissions for California. How representative is CA of the rest of the US?

We believe that the added detail in the introduction described above will better connect this section with the rest of the manuscript. We've shown the motivation of this work is to help unambiguously identify VOC tracer compounds linked to coatings. We do this through ambient and laboratory measurements, while the third section shows that Texanol and PCBTF are VOCs unique to the coatings category and their emissions are prevalent in inventories.

The reason this section focuses on California is based solely on the availability of data. California is the only state that conducts comprehensive surveys of coatings as detailed in the text. It is likely the product sales and usage in California are similar throughout North America. The FIVE-VCP inventory that

includes data from these surveys has shown good agreement with ambient measurements in LA and NYC (McDonald et al., 2018; Coggon et al., 2021).

Referee #3

VCPs' emissions have become a very important consideration in tropospheric chemistry studies, especially those focused on developed megacity environments. It is now essential to determine the major contributors to these emissions and specific tracer species that could assist in their isolation and quantification. From this perspective, this paper certainly would be useful for the scientific community. The authors have specifically focused on emissions from coatings-related products and have approached the source from multiple investigative dimensions including laboratory investigations, ambient measurements and exploration of existing chemical speciation surveys and databases. The work is comprehensive in nature and attempts to encompass all manners of arguments to support the paper's objectives. Nevertheless, I have some concerns that need to be resolved before I could recommend the work for publication. They are listed below:

1. The introduction mostly focuses on VOCs which for the large part is fair for coatings related products. Still, to present the complete picture, I believe some acknowledgement is warranted for small IVOCs (n-alkane equivalent volatility< C14) that may be present in solvents-based coating products. The authors do lend a quick word to LVPVOCs that are currently exempt, but others have shown that LVP-VOCs are important for air quality and yet maybe underestimated due to instrumentation limitations and long emission timescales e.g. Khare and Gentner, 2018 ACP.

We agree that we should mention I/SVOCs in the same context that we describe LVP-VOCs even though our instrumentation primarily measures VOCs and only a few IVOCs. We add text to the introduction to highlight the importance of these understudied SOA precursors and their exclusion from inventories and regulations.

"Compounds classified as intermediate-volatility and semi-volatile organic compounds (IVOCs and SVOCs) are often excluded from inventories and regulations due to measurement limitations or long emission time-scales, yet many are key SOA precursors that can be emitted by coating products or processes (Khare and Gentner, 2018)."

Added to references: "Khare, P. and Gentner, D. R.: Considering the future of anthropogenic gas-phase organic compound emissions and the increasing influence of non-combustion sources on urban air quality, Atmos. Chem. Phys., 18, 5391–5413, https://doi.org/10.5194/acp-18-5391-2018, 2018."

2. The experimental methods section could be better knitted. For example, as a reader, I'm confused about why I am reading 2.1.1 and 2.1.2 separately. As I understand it, 2.1.2 involves coupling a GC with PTR-ToF ahead of its inlet for improved separation that also permits detection of isomers. This could be easily merged with 2.1.1 and narrated cohesively.

The experimental section is broken down into four parts: (1) description of the instrumentation (2) description of the laboratory measurements (3) description of ambient measurements and (4) description of the FIVE-VCP inventory. Based on this and later suggestions we have reorganize the experimental methods section to mirror the discussion (i.e., ambient measurements, FIVE-VCP inventory used with the ambient measurements, then laboratory measurements) and combine 2.1.1 and 2.1.2 together as suggested.

3. Similarly, in laboratory measurements subsection (in methods), I would like to see some description (with perhaps a schematic in the SI) of the headspace sampling setup in the main text. Future experimental studies on VCPs could certainly benefit from it. The current description of laboratory methods left me with questions: Was the sample well-mixed (or re-mixed) within the container before headspace sampling occurred? Was the sample transferred to a new container for headspace sampling or

was it done in the original product container? If the sample was transferred to a new container, was it collected from the core of the original product or scooped from the surface since the core may have more volatile content? What temperature was the product subjected to during emissions? How much time was allowed for air-sample equilibrium to be attained in the headspace? I understand that this was a qualitative investigation but the finer details are important to ascertain that the sample was not already partially depleted of certain volatile constituents before sampling was conducted.

We thank the reviewer for their suggestion to give greater detail on the headspace experiments and we hope the added detail will benefit future experimentalists. We add text to the main manuscript describing the sampling approach, although we believe the detailed description provides enough context that it can readily be simulated in future studies without the addition of a schematic. As a brief description, the products were mixed thoroughly and an aliquot was transferred to a new container. The container was opened to the atmosphere at room temperature and GC samples were collected every 20 minutes, resealing the container between runs. As you mentioned, these were qualitative experiments, however, the products were not allowed to dry and repeat GC runs revealed repeated peaks.

Added text: "Each product was mixed thoroughly in its original container and an aliquot was transferred into a glass vial. The headspace VOCs were sampled by placing the product container that was open to the atmosphere within a few centimeters of the PTR-ToF inlet. The instrument inlet tubing was a short piece of PFA Teflon to limit inlet losses or delays (Deming et al., 2019). The container was closed between GC sample intervals (20 min) and zero air was sampled between runs."

Added to references: "Deming, B. L., Pagonis, D., Liu, X., Day, D. A., Talukdar, R., Krechmer, J. E., de Gouw, J. A., Jimenez, J. L., and Ziemann, P. J.: Measurements of delays of gas-phase compounds in a wide variety of tubing materials due to gas–wall interactions, Atmos. Meas. Tech., 12, 3453–3461, https://doi.org/10.5194/amt-12-3453-2019, 2019."

4. Lines 131-132: Please define extreme environmental conditions. Non-combustion emissions such as those from coatings would be temperature-dependent. Could excluding products used in high-temperature environments affect/bias your results in any way, especially the emission rates?

Industrial maintenance coatings are defined as high performance architectural coatings applied to substrates formulated for extreme environmental conditions such as: (a) Immersion in water, wastewater, or chemical solutions or chronic exposure of interior surfaces to moisture condensation; (b) Acute or chronic exposure to corrosive, caustic, or acidic agents, or similar chemicals, chemical fumes, chemical mixtures, or solutions; (c) repeated exposure to temperatures above 121 C; (d) Frequent heavy abrasion, including mechanical wear and frequent scrubbing with industrial solvents, cleansers, or scouring agents; (e) Exterior exposure of metal structures. The focus of this study was on readily-available coatings products and most commercially available products are formulated to withstand some degree of environmental exposure. There is significant variability in coatings products, and only a subset of products was selected for this analysis and we do not recommend the emission rates calculated are representative of all coatings. Each individual product, whether it be formulated for extreme or normal conditions, has its unique composition and the aim of this study was to identify compounds prevalent across several products supporting their use as tracers and to evaluate emissions after initial application at room temperature conditions. It is certainly possible the formulations for the products designed for extreme conditions might not include PCBTF and Texanol and some products likely don't include these compounds as ingredients. At the suggestion of Referee #2 we've added text to the Laboratory Measurement section noting product variability with a disclaimer that these products were selected based on availability, VOC content, and chemical similarity to other tested products.

5. Lines 132-133: It is unclear how this was positioned. Was the PTR-ToF inlet exposed to an open product container? What was the distance between the inlet and the product? Was the direction of the airflow between the inlet and sample controlled? The issues go with point 3 above.

We've addressed these questions by expanding on the description of the headspace experiments in the experimental section as suggested in point 3 above.

6. Lines 138-140,309: Please mention what was the range of the emissions velocity from the product's surface into the air flow. For quantification purposes, it is important that the emissions velocity is representative of environmental conditions in which the product is to be used. Otherwise, please provide an explanation for how the measured emission rates could be justified as relevant in real-world conditions, and also from a modeling perspective. What temperature was each sample subjected to during the chamber experiments?

As requested earlier, we have added to the experimental details that the evaporation experiments were all conducted at room temperature. An emissions velocity from the product's surface was not measured, instead a constant flow of synthetic zero air continuously passed through the Teflon coated chamber as noted in the Laboratory Measurements section. Only a subset of products was selected for this analysis and we do not recommend the emission rates calculated are representative of all coatings. The emission rates qualitatively highlight the differences between product types with varying VOC content and demonstrate the time-dependent variability following application.

7. It would probably help to keep the discussion sections consistent with the experimental subsections or vice versa. As a reader, I wasn't sure why I landed on a discussion section specifically focused on Texanol and PCBTF right after the methods. It might be better to present the bigger picture, broader details about the chemical speciation of tested products and field measurements, before narrowing it down to the proposed tracer compounds.

We note that the order of the Experimental Methods did not mirror the discussion text and we have now reordered the methods section to give a better flow between the two sections. Additionally, based on the suggestion of Referee #2 we added text to the introduction that highlights the reasoning for the order of the manuscript since the focus of this study is to help unambiguously identify VOC tracer compounds linked to coatings observed in ambient measurements. This rationalizes the structure of the manuscript starting with initial ambient observations of two potential tracers, followed by confirming their prevalence in several coating products, and finally evaluating their uniqueness and usage in inventories.

8. Section 4 "Headspace and Evaporation Measurements" could be written better. The paragraphs read disjointed and I had to scroll up and down repeatedly to properly understand the content. For example, line 244 marks initiation of the discussion of headspace sampling which in line 268 suddenly shifts to evaporation experiments without any ramp.

To make the section read clearer we've added a sentence to show a separation between the discussion of headspace analyses and evaporation experiments. The headspace discussion already begins with the following sentence: "To understand the composition of the emissions from coating products in greater detail, the vapors from the headspace of nineteen different coatings were measured by PTR-ToF." To smoothly transition to the discussion focused on evaporation experiments, the introduction to the paragraph at L268 is reordered and modified as follows: "During evaporation experiments, the products were introduced into an enclosed chamber and sampled over longer periods to investigate changes in evaporative emissions." The subsequent paragraphs describe results from the evaporation experiments, however, there is still considerable overlap between the headspace and evaporation experiments when discussing the dominant emissions from each product. The main text explicitly references GC-chromatograms or the GC analysis when detailing those results.

9. Lines 186-187: If PCBTF and Texanol are predominantly from construction activities, some explanation is required for why their mixing ratios were dominated by short, isolated plumes? I would expect construction activity in an area to be decaying but continuous source of the tracer, especially if the source is industrial coatings. Or is this because the mobile lab was in continuous movement and saw a temporary spike when it happened to be passing by a source? I think it is the latter based on the opening statements in the conclusion paragraph. However, it is important to clarify this, or else source behavior could be easily misunderstood.

It is likely the confusion results from the first sentence of the Ambient Measurements of PCBTF and Texanol. We will remove "ground site" from the first sentence as the initial discussion focuses on mobile measurements and ground site measurements aren't considered until temporal trends are discussed. This should alleviate the confusion.

10. Lines 349-351: The authors mention that acetone and formaldehyde did not emit as expected based on thermodynamic considerations but leave it at that calling it "complicated". I found the trends in Figure 7b very interesting and would very much like to see some explanation, even if just reasonable speculation from the authors for these observed trends.

It is very likely fragmentation plays an important role here, with significant fragmentation to both these masses this complicates using saturation vapor concentrations to predict peak emission. There are also other variables likely in play as described in the text (e.g., composition, drying time, substrate interaction/properties, etc.)

Added text: "Fragmentation of larger compounds with a range of volatilities, including glycols, to the *m/z* of acetone and formaldehyde is one possible explanation for these discrepancies"

11. Figure 2b could go into the SI. Since these are mobile measurements, the concentration measured is not singularly dependent on time of day but also the location of the vehicle. Hence, the figure is confusing and not substantial enough in my perspective to warrant a spot in the main text. Would the highest peaks still be observed between 2-3 PM if the vehicle was at some other location? The figure caption should clearly mention that these are mobile laboratory-based ambient measurements.

Figure 2b was initially included to further emphasize that PCBTF shows hot spot, episodic behavior. However, we agree with the Referee that the figure does not add significantly to the manuscript and Fig. 2a clearly shows mobile measurements of spatially, isolated plumes. We have removed Figure 2b.

Minor points: -Line 244-245: Nitpicking here but measuring the headspace by PTR-ToF is unclear. Sampled vapors from the headspace is more accurate. Corrected

 -Fig. vs. Figure is used inconsistently in the text.Line 268-"Figure 5a", line 277-"Fig. 5b". Similarly for 6. Line 310 says "Figure 6", line 314- "Fig. 6". Please check others.

The manuscript preparation guidelines outlined by the manuscripts state: "The abbreviation "Fig." should be used when it appears in running text and should be followed by a number unless it comes at the beginning of a sentence, e.g.: "The results are depicted in Fig. 5. Figure 9 reveals that...".

-Line 392: increased "from". Corrected

-Figure 3: (a) has illegible overlap in the figure. –

There is very limited overlap for the labeled points. We have eliminated the second Chicago from the top of panel (a) to eliminate the most significant overlap

Figure 5: is clumsy. Compound identifications overlap and are not readable in several places. Consider using arrows where necessary. Some compound names are not fully printed in the spectra. -All across the figures, the legends sometimes start with block letters and other times with small letters. Please correct the inconsistency.

We have limited the number of species labeled in the figure and started each species name with capital letters to clean up the image. The main text is now changed from "The major peaks in each product are labeled" to "Select major peaks in each product are labeled"

Voluntary Changes: We have updated the reference of Gkatzelis et al. (2020a) to Gkatzelis et al. (2021a) as it is now published. We have also updated to the current year (2021) for manuscripts currently submitted for publication (i.e., Coggon et al. 2020 and Gkatzelis et al. 2020b)

Gkatzelis, G. I, Coggon, M. M., McDonald, B. C., Peischl, J., Aiken, K. C., Gilman, J. B., Trainer, M., and Warneke, C.: Identifying volatile chemical product tracer compounds in U.S. cities, Environ. Sci. Technol., submitted, 55 (1), 188-199, doi: 10.1021/acs.est.0c0546, 2021a.

---

## Author Response (AR2)

Response to Editor:

We thank the referees and editor for their helpful reviews. The manuscript has been revised accordingly with our responses interspersed below.

**There are still a few remaining issues from the reviewers. Please address or clarify accordingly. Thanks.**

**(1) Figure 3b and associated text in lines 232-248: The ratio of the paint tracers to benzene is not systematically different on weekends compared to weekdays. What is the trend in concentrations? My assumption would be that benzene concentrations are similar on weekends to weekdays because automobile traffic typically has a small reduction on the weekends (compared to diesel, which falls significantly). But perhaps NYC is atypical in this regard and there are large reductions in vehicle traffic on the weekend.**

We highlighted in the text that we might expect the coatings emissions to be smaller on the weekends compared to weekdays if industrial and/or construction activities reduced on the weekend, but this trend was not observed as shown in Figure 3b. It may even be true that these activities persist into the weekend, though data on coatings usage in NYC does not currently exist. As the reviewer suggested, the benzene concentrations only had a slight reduction from the weekdays to weekend days, with the lowest observed benzene concentration reported on Sunday. Because weekend benzene reductions were not significant, the ratios of the tracer compounds (PCBTF and Toluene) to benzene reflect changes in the usage patterns of coatings products that use these compounds as ingredients from weekdays to weekend days. The results suggest coating usage might not change by day of week in densely populated New York City, however, these usage trends might not reflect emissions in other U.S. cities. In order to draw more robust conclusions about weekend/weekday coatings usage, ambient measurements in other cities would be necessary. To better clarify these conclusions, the following text has been modified at L246:

"The mean concentration of benzene does not decrease significantly from weekdays to weekend days, therefore changes in the emission ratios to benzene by day of week should reflect changes in the usage of coatings products that use PCBTF and Texanol as ingredients. It might also be expected that coatings emissions are smaller on the weekends compared to weekdays, but this difference is not clearly observed in Fig. 3b and alternatively suggests that coating usage persists into the weekend in New York City, though these trends likely vary by city."

**(2) Figure 5 pie charts - for the SB polyurethane, it seems that aromatics are grouped into "HC". Please clarify.**

The pie charts in Figure 5b show the distribution of VOC mass emitted and measured during each complete evaporation experiment. The largest 10 masses measured by the PTR-ToF-MS are labeled as the "most-likely" species. The PTR-ToF-MS measures the chemical formulae; however, isomers cannot be distinguished and explicit identification is not always possible. This

is especially true at low *m/z*, where large hydrocarbons (>C8) fragment complicating spectra interpretation (Erickson et al 2014; Gueneron et al., 2015). For the solvent-borne polyurethane, the pie chart is labeled based on the formula at the detected mass (i.e. isoprene, cyclopentane fragments, and other fragments labeled as C4 HC). The explicit identity of such a complex mixture of hydrocarbons that uses Stoddard solvent as its base is not possible using PTR-ToF-MS as the primary detection technique. Application of the GC as the front-end of the PTR even highlights the complexity of the mix of hydrocarbons in this product (alkanes, alkenes, cycloalkanes, aromatics, etc.). It is possible that aromatic fragments are included at some of the lower *m/z*, however, explicit identification is not possible. In order to clarify this, the following text is added to the Figure 5 caption:

"Several small hydrocarbon masses could not be explicitly identified due to fragmentation of larger masses and are instead labeled by their carbon number."

Erickson, M. H., Gueneron, M., and Jobson, B. T.: Measuring long chain alkanes in diesel engine exhaust by thermal desorption PTR-MS, Atmos. Meas. Tech., 7, 225–239, doi:10.5194/amt-7-225-2014, 2014.

Gueneron, M., Erickson, M. H., VanderSchelden, G. S., and Jobson, B. T.: PTR-MS fragmentation patterns of gasoline hydrocarbons, Int. J. Mass Spectrom., 379, 97–109, doi:10.1016/j.ijms.2015.01.001, 2015.

**(3) Figure 8a: A few sentences on how the CARB inventory calculates emissions would be warranted. Presumably the inventory emissions reflect the changing composition of paints and coatings over the period of 1990-2014.**

CARB utilizes sales, formulation, and reactivity data from mandatory surveys conducted every few years. These data were used to calculate the quantity of total organic gases and total reactive gases (VOCs and LVP-VOCs). CARB then applies a fate and transport adjustment to ingredients to correct for other loss processes including down the drain or combustion as determined by a comprehensive review of the current literature. The data included in Figure 8a, contains survey data from five separate years (1990, 1998, 2001, 2005, and 2014) and reflects both the changing composition and usage (from sales) of coatings in California. L413-418 gave a brief description of the CARB inventory, however, we have re-worked the introductory paragraph of that section (L408) to improve the details of these emissions calculations:

"Every four to five years, CARB conducts comprehensive surveys of architectural coatings sold in California to gather information about the ingredients and sales with the goal of updating emission inventories. The response to the surveys is mandatory and CARB ensures the validity of the data following extensive quality assurance and quality control measures and the results accurately represent the sales volume in California and are made publicly available (CARB, 2018). These sales, product formulations, and reactivity data are combined together with fate and transport adjustments to account for non-atmospheric loss processes to estimate the total emissions of reactive organic gases."

**(4) It seems that one impact of this work is that it informs how paint and coating emissions can be represented in chemical transport models. Since most of the VOCs seem to be emitted very quickly (a few hours), does it mean that models can ignore longer-term emissions that happen over multiple days?**

We do note that the majority of the VOC mass measured in these evaporation experiments was emitted within a few hours of application, however, the emissions and the timescale of emissions are also driven by temperature, film thickness, substrate material, and compound volatility. This study only sampled a few product formulations in an enclosed system at room temperature on the same substrate and did not systematically investigate temperature dependencies (Khare et al., 2020), application variability (Zhao et al., 2020), or long-term emissions (Lin and Corsi, 2007). Additionally, the PTR-ToF-MS is sensitive to many VOCs, though it is not sensitive to many larger, highly functionalized, lower volatility compounds traditionally categorized as intermediate and semi-volatile organic compounds that have long emission time-scales and have been shown to be emitted by coating products or processes (Khare and Gentner, 2018). These lower volatility compounds are likely key SOA-precursors in some regions and would need to be included in chemical transport models. While the VOCs measured by the PTR-ToF-MS could contribute to initial ozone formation following application, the changes in the emissions over time are still not well understood. The main result of this study is that VOC emissions from coatings, specifically PCBTF and Texanol, can be used as tracers for application events and usage of coatings products.

We were careful to highlight that only VOCs were measured in this study and L364-366 notes these emissions likely contribute to ozone formation, but we do not comment on any potential contributions to SOA formation. We also noted at L75-78 that SVOCs and IVOCs are key SOA precursors from coatings and have traditionally been excluded from models due to challenges in detection and quantification. We have modified the text at L364-366 to better emphasize the limitations of this study:

"For the coatings tested here, the majority of the measured VOC mass is emitted within the first few hours, and therefore the most significant atmospheric implications for ozone formation likely occur during and shortly following application, though the changes in the VOC emissions over longer timescales and the effects of environmental and application parameters were not systematically investigated in this study."

Khare, P., Machesky, J., Soto, R., He, M., Presto, A. A., and Gentner, D. R: Asphalt-related emissions are a major missing nontraditional source of secondary organic aerosol precursors, Science Advances, 6(36), DOI: 10.1126/sciadv.abb9785, 2020.

Zhou, X., Gao, Z., Wang, X., and Wang, F.: Mathematical model for characterizing the full process of volatile organic compound emissions from paint film coating on porous substrates, Building and Environment, 182, doi: 10.1016/j.buildenv.2020.107062, 2020.